# Generalized Linear Mode Connectivity for Transformers

**Alexander Theus** [1 2]
atheus@ethz.ch

**Alessandro Cabodi** [1]
acabodi@ethz.ch

**Sotiris Anagnostidis** [1]
sanagnos@ethz.ch

**Antonio Orvieto** [3 4 5]
antonio@tue.ellis.eu

**Sidak Pal Singh** [*1 2]
contact@sidakpal.com

**Valentina Boeva** [*1 6 7]
vboeva@ethz.ch

## Abstract

Understanding the geometry of neural network loss landscapes is a central question in deep learning, with implications for generalization and optimization. A striking phenomenon is *linear mode connectivity* (LMC), where independently trained models can be connected by low- or zero-barrier paths, despite appearing to lie in separate loss basins. However, this is often obscured by symmetries in parameter space—such as neuron permutations—which make functionally equivalent models appear dissimilar. Prior work has predominantly focused on neuron reordering through permutations, but such approaches are limited in scope and fail to capture the richer symmetries exhibited by modern architectures such as Transformers. In this work, we introduce a unified framework that captures four symmetry classes—permutations, semi-permutations, orthogonal transformations, and general invertible maps—broadening the set of valid reparameterizations and subsuming many previous approaches as special cases. Crucially, this generalization enables, for the first time, the discovery of low- and zero-barrier linear interpolation paths between independently trained Vision Transformers and GPT-2 models. Furthermore, our framework extends beyond pairwise alignment, to multi-model and width-heterogeneous settings, enabling alignment across architectures of different sizes. These results reveal deeper structure in the loss landscape and underscore the importance of symmetry-aware analysis for understanding model space geometry. Our code is available here.

## 1   Introduction

Understanding the geometry of neural network loss landscapes is central to both theoretical and practical advances in deep learning. A key observation driving this line of research is that independently trained models, despite converging to different points in parameter space, can sometimes be connected by low-loss paths, suggesting an unexpected interrelation between seemingly isolated loss minima [Freeman and Bruna, 2016, Garipov et al., 2018, Tatro et al., 2020]. When these connecting paths are approximately linear and maintain low loss throughout, the phenomenon is known as *Linear Mode Connectivity* (LMC) [Frankle et al., 2020, Entezari et al., 2021].

---

[*] Equal advising.
[1]ETH Zürich, [2]MPI for Learning Systems, [3]ELLIS Institute Tübingen, [4]MPI for Intelligent Systems, [5]Tübingen AI Center, [6]Swiss Institute of Bioinformatics, [7]Université Paris Cité, Institut Cochin, INSERM U1016

39th Conference on Neural Information Processing Systems (NeurIPS 2025).

LMC challenges the naive view of loss landscapes as consisting of distinct basins separated by high barriers. Instead, it hints at a reparameterization-induced redundancy: multiple minima that appear distant in weight space may correspond to functionally similar solutions, made to appear dissimilar due to symmetries in the parameterization. Recovering LMC between models thus requires aligning their parameters into symmetry-equivalent configurations that reveal their underlying functional equivalence.

Several state-of-the-art approaches leverage discrete neuron permutations to achieve such alignments, demonstrating LMC for relatively simple architectures like shallow multilayer perceptrons (MLPs) and, under certain conditions, VGG networks and ResNets [Singh and Jaggi, 2020, Ainsworth et al., 2022]. Such works highlight the central role of symmetries in understanding neural network landscapes and inspired the development of so-called model *re-basin* techniques: transformations that port independently trained networks into a common valley of the loss landscape.

Although foundational, permutation symmetries alone do not capture the full range of symmetries exhibited by modern architectures such as Transformers. Our empirical findings show that relying solely on discrete reordering often fails to reveal low-loss paths, as persistent barriers can remain even after permutation alignment [Verma and Elbayad, 2024].

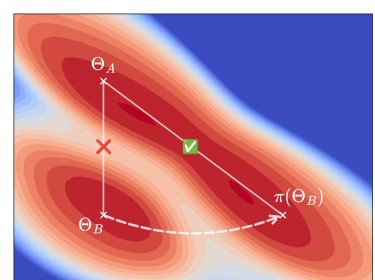

Figure 1: By considering network symmetries beyond permutations, we can teleport two independently trained Transformers to the same loss basin. $\Theta_B$ is projected into a functionally equivalent representation $\pi(\Theta_B)$.

In this work, we broaden the symmetry lens. We introduce a unified framework that formalizes four classes of transformations: permutations, semi-permutations, orthogonal transformations, and general invertible maps which, if appropriately used, can produce valid reparameterizations under which the original model functionality is preserved, while achieving low barriers. This generalization subsumes several existing alignment techniques as special cases and provides the means to uncover LMC in Transformers by allowing to identify and leverage their richer symmetry classes. Furthermore, this formalization also accommodates alignment between heterogeneous Transformers with differing architectural widths.

Our empirical results demonstrate for the first time low- and zero-loss linear connections between independently trained Vision Transformers and GPT-2 models, even in multi-model settings. These findings suggest that the Transformers' loss landscape is more connected than previously thought, provided the symmetries at play are adequately modeled and exploited (see Figure 1). We discuss broader implications for ensembling, federated and continual learning, adversarial robustness, and the role of positional encodings in Appendix F.

This paper advances the understanding of loss landscape geometry and model alignment through the following key contributions:

i **Unified symmetry framework:** We formalize a broad class of parameter transformations — including permutations, semi-permutations, orthogonal transformations, and general invertible maps — that preserve model functionality. This unifies and extends prior approaches under a single theoretical lens and enables alignment across both homogeneous and heterogeneous architectures (Section 3).

ii **LMC for Transformers:** We demonstrate, for the first time, low- and zero-loss linear paths between independently trained Vision Transformers and GPT-2 models using richer symmetry classes (Section 4 and 5).

iii **Multi-model mode connectivity:** Our framework extends to the multi-model setting, revealing that several independently trained transformers can be merged while maintaining a near-zero interpolation barrier (Section 4 and 5).

iv **Soft alignment via continuous symmetries:** We show that relaxing exact equivalence through differentiable, non-discrete transformations may improve interpolation outcomes (Appendix D).

## 2 Generalizing Linear Mode Connectivity

Let $\boldsymbol{\theta}$ be network parameters and $f[\boldsymbol{\theta}](\cdot)$ the induced function. Let $\ell[\boldsymbol{\theta}](x, y)$ be the loss incurred by $f[\boldsymbol{\theta}]$ on a data point $(x, y)$. For a dataset $\mathcal{D} = \{(x_i, y_i)\}_{i=1}^N$, define the empirical risk

$$\mathcal{L}[\boldsymbol{\theta}](\mathcal{D}) = \frac{1}{N} \sum_{i=1}^N \ell[\boldsymbol{\theta}](x_i, y_i).$$

**Definition (Linear Mode Connectivity).** Consider two models $\boldsymbol{\theta}_A, \boldsymbol{\theta}_B$, both pre-trained on the same task (for simplicity). They are *linearly mode connected* if $\mathcal{L}[\boldsymbol{\theta}_A](\mathcal{D}) \simeq \mathcal{L}[\boldsymbol{\theta}_B](\mathcal{D})$ and the interpolation barrier

$$\mathcal{B}_\lambda[\boldsymbol{\theta}_A, \boldsymbol{\theta}_B](\mathcal{D}) = \mathcal{L}[\lambda \boldsymbol{\theta}_A + (1 - \lambda)\boldsymbol{\theta}_B](\mathcal{D}) - \left(\lambda \mathcal{L}[\boldsymbol{\theta}_A](\mathcal{D}) + (1 - \lambda) \mathcal{L}[\boldsymbol{\theta}_B](\mathcal{D})\right)$$

is near-zero for all $\lambda \in [0, 1]$. The empirical barrier is:

$$\mathcal{B}[\boldsymbol{\theta}_A, \boldsymbol{\theta}_B](\mathcal{D}) = \sup_{\lambda \in [0,1]} \mathcal{B}_\lambda[\boldsymbol{\theta}_A, \boldsymbol{\theta}_B](\mathcal{D}), \tag{1}$$

and LMC is observed when $\mathcal{B}[\boldsymbol{\theta}_A, \boldsymbol{\theta}_B](\mathcal{D}) \approx 0$.

Because neural networks admit non-unique parameterizations [Li et al., 2023], especially under different initializations, we seek invertible, function-preserving alignment mappings $\pi : \Theta \to \Theta$ that, when applied to $\boldsymbol{\theta}_A$ and $\boldsymbol{\theta}_B$, encourage LMC:

$$\min_{\pi_A, \pi_B} \quad \mathcal{B}[\pi_A(\boldsymbol{\theta}_A), \pi_B(\boldsymbol{\theta}_B)](\mathcal{D}) \quad \text{s.t.} \quad f[\pi_A(\boldsymbol{\theta}_A)] = f[\boldsymbol{\theta}_A], \ f[\pi_B(\boldsymbol{\theta}_B)] = f[\boldsymbol{\theta}_B].$$

These mappings need not be mere permutations. As discussed in Section 3, modern architectures (e.g., Transformers) exhibit richer, component-specific symmetries that can more effectively reduce—or eliminate—the barrier (Appendix E). This broader view is not confined to networks with identical architecture and naturally extends to enable alignment across models of differing widths.

**Extension to multi-model connectivity.** For $M$ independently trained models $\{\boldsymbol{\theta}_m\}_{m=1}^M$, define the *multi-model barrier*

$$\mathcal{B}[\{\boldsymbol{\theta}_m\}_{m=1}^M](\mathcal{D}) = \sup_{\boldsymbol{\lambda} \in \Delta^{M-1}} \left[\mathcal{L}\Big[\sum_{m=1}^M \lambda_m \boldsymbol{\theta}_m\Big](\mathcal{D}) - \sum_{m=1}^M \lambda_m \mathcal{L}[\boldsymbol{\theta}_m](\mathcal{D})\right], \tag{2}$$

where $\Delta^{M-1} = \{\boldsymbol{\lambda} \in \mathbb{R}_{\geq 0}^M : \sum_m \lambda_m = 1\}$. Multi-model linear connectivity holds when $\mathcal{B}[\{\boldsymbol{\theta}_m\}](\mathcal{D}) \approx 0$, indicating a shared low-loss basin. As in the pairwise case, we search for symmetry-preserving mappings $\{\pi_m\}_{m=1}^M$ satisfying $f[\pi_m(\boldsymbol{\theta}_m)] = f[\boldsymbol{\theta}_m]$ that minimize $\mathcal{B}[\{\pi_m(\boldsymbol{\theta}_m)\}](\mathcal{D})$, forming the basis for the merging procedure in Section 4.3.

## 3 Network symmetries under the generalized framework

To perform alignment and enable meaningful interpolation between independently trained models, we must first understand the underlying symmetries that govern neural network parameter spaces. While prior work has focused primarily on discrete permutations, these represent only a slice of the broader symmetry landscape. In this section, we introduce a hierarchy of network symmetries—*permutation*, *semi-permutation*, *orthogonal*, and *invertible*—each allowing progressively more flexible function-preserving transformations, as summarized in Table 1. We define each class, identify where it arises in neural architectures, and conclude by showing how these symmetries manifest in Transformer models, enabling their effective alignment and merging. More theoretical grounding and formal analysis of these symmetry classes can be found in Appendix E.

### 3.1 Symmetry classes

**Permutation.** Permutation symmetry refers to transformations that reorder inputs while preserving the network's function. It arises when components treat each input dimension independently, allowing neuron reordering without affecting the output. This symmetry is characteristic of elementwise operations such as GELU, sigmoid, softmax, tanh, where output values correspond directly to input positions.

Table 1: Hierarchical organization of symmetry classes in neural network components, illustrating their associated transformation structures and representative examples. Each class is a strict subset of the one below it.

| Hierarchy | Class | Structure | Examples |
|:---:|:---|:---:|:---:|
| $\mathcal{S}_1$ | Permutation | Permutation matrices ($\mathbf{P}$) | GELU, sigmoid, softmax, $\tanh$ |
| $\subset \mathcal{S}_2$ | Semi-permutation | Sparse, stochastic matrices ($\tilde{\mathbf{P}}$) | RELU, LayerNorm, MHA |
| $\subset \mathcal{S}_3$ | Orthogonal | Orthogonal matrices ($\mathbf{O}$) | RMSNorm |
| $\subset \mathcal{S}_4$ | Invertible | Full-rank matrices | Linear layer |

**Semi-permutation.** "Semi-permutation" symmetry extends permutation symmetry by allowing sparse, weighted mixing of input dimensions. It is defined by matrices $\mathbf{P} \in \mathbb{R}^{M \times N}$, where $M \geq N$, each column is a stochastic vector, and each row contains at most one positive entry. This symmetry arises in components that are *linearly decomposable*—that is, their functional output satisfies the following identity:

$$f(x) = f(\alpha x) + f((1-\alpha)x), \quad \forall \alpha \in [0, 1],$$

which holds for piecewise-linear functions such as RELU, PRELU, and the absolute value. These functions permit structured, non-permutative mixing of input channels while preserving functionality. As permutations are a subset of semi-permutations, many existing works focus on permutation symmetries of RELU based activation networks.

**Orthogonal.** Orthogonal symmetry allows transformations that preserve vector norms and angles—such as rotations and reflections—without altering the network's behavior. This symmetry arises in components that normalize inputs across dimensions, irrespective of their orientation in space. *RMSNorm* is a key example, remaining invariant under orthogonal transformations.

**Invertible.** Neural network components that preserve linearity admit functional equivalence under invertible transformations — the most general symmetry class in our hierarchy. This class includes layers whose learned transformations are unconstrained by structural restrictions such as sparsity or orthogonality. A key example is the *attention mechanism*, where the *QK* and *OV* circuits [Elhage et al., 2021] (i.e., the projection weights for queries, keys, and values) can be reparameterized via invertible maps without altering model behavior.

**Approximate invariance and soft symmetries.** The strict requirement of functional equivalence $f[\boldsymbol{\theta}'](\mathbf{X}) = f[\boldsymbol{\theta}](\mathbf{X})$ can be relaxed to approximate equality $f[\boldsymbol{\theta}'](\mathbf{X}) \approx f[\boldsymbol{\theta}](\mathbf{X})$, allowing continuous transformations to serve as soft symmetry operations. In this setting, soft permutations are represented by doubly stochastic matrices, computed via entropic optimal transport or Sinkhorn-based projections. Such relaxations enable more flexible neuron alignments—e.g., many-to-one or one-to-many mappings— and can improve test-time performance (see Appendix D).

### 3.2 Symmetries in Transformers

#### 3.2.1 Feed-forward layer

The Transformer's feed-forward layer applies two linear projections with a nonlinearity in between:

$$\text{FF}(\mathbf{x}) = \mathbf{W}_2\,\phi(\mathbf{W}_1\mathbf{x} + \mathbf{b}_1) + \mathbf{b}_2,$$

where $\phi$ is an elementwise activation function, typically GELU or RELU. When $\phi$ is not linearly decomposable—such as GELU—the layer only admits a strict *permutation symmetry*. If $\phi$ instead is piecewise linear (e.g., RELU), the layer also admits a broader *semi-permutation symmetry* as described in Section 3.1. For any permutation matrix $\mathbf{P}_{\text{FF}} \in \mathbb{R}^{h \times h}$, the reparameterization

$$\mathbf{W}_1' = \mathbf{P}_{\text{FF}}\mathbf{W}_1, \quad \mathbf{W}_2' = \mathbf{W}_2\mathbf{P}_{\text{FF}}^\top, \quad \mathbf{b}_1' = \mathbf{P}_{\text{FF}}\mathbf{b}_1$$

yields an equivalent function:

$$\text{FF}'(\mathbf{x}) = \text{FF}(\mathbf{x}).$$

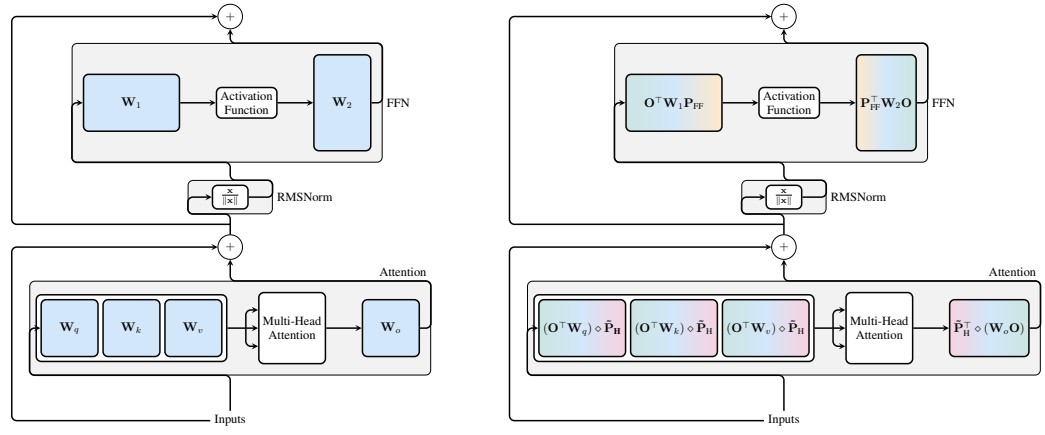

(a) Transformer layer.  (b) Transformer layer after projection.

Figure 2: (a) illustrates a standard Transformer layer, and (b) shows its function-preserving equivalent after structured weight transformations. Attention heads are semi-permuted via blockwise multiplication with $\tilde{\mathbf{P}}_\mathbf{H}$ using the $\diamond$ operator, feedforward weights are permuted via $\mathbf{P}_\mathbf{FF}$, and residual stream weights are orthogonally transformed via $\mathbf{O}$. The figures are inspired by Ashkboos et al. [2024].

### 3.2.2 Multi-head attention

The multi-head attention mechanism of Transformers is defined as:

$$\text{MultiHead}(\mathbf{Q}, \mathbf{K}, \mathbf{V}) = \sum_{i=1}^{H} \underbrace{\text{softmax}\left(\frac{(\mathbf{Q}\mathbf{W}_i^Q)(\mathbf{K}\mathbf{W}_i^K)^\top}{\sqrt{d_k}}\right)(\mathbf{V}\mathbf{W}_i^V)}_{\text{head}_i(\mathbf{Q}, \mathbf{K}, \mathbf{V})} \mathbf{W}_i^O.$$

**Intra-head.** Each attention head exhibits two *invertible symmetries*, as formally characterized in the QK and OV circuits by Elhage et al. [2021]. The first arises from the product $\boldsymbol{W}_i^Q(\boldsymbol{W}_i^K)^\top$, which governs the attention scores (QK-circuit). The second appears in $\boldsymbol{W}_i^V\boldsymbol{W}_i^O$, which maps values to outputs (OV-circuit). Because any invertible transformation applied within the query, key, value, or output projections can be algebraically multiplied out, the behavior of the attention mechanism is uniquely determined by these two products. Consequently, we omit explicit invertible reparameterizations and treat the QK and OV circuits as canonical representations of each head.

**Inter-head.** Multi-head attention exhibits a *semi-permutation symmetry* across heads. Since heads are summed independently, their order is irrelevant (permutation symmetry). Moreover, heads can be decomposed linearly:

$$\text{head}(\mathbf{X}; \mathbf{QK}, \mathbf{OV}) = \text{head}(\mathbf{X}; \mathbf{QK}, \alpha \cdot \mathbf{OV}) + \text{head}(\mathbf{X}; \mathbf{QK}, (1 - \alpha) \cdot \mathbf{OV}),$$

for any $\alpha \in \mathbb{R}$. This enables structured reweighting and mixing of heads via sparse, stochastic matrices $\tilde{\mathbf{P}}_\mathbf{H}$, placing multi-head attention in the semi-permutation class.

### 3.2.3 Residual

Recently, Ashkboos et al. [2024] observed that Transformers exhibit an orthogonal symmetry along the residual path, where the only non-linear component is the normalization layer. When RMSNorm is employed, the model exhibits orthogonal symmetry directly (see Section 3.1). In contrast, when LayerNorm is used, it can be reformulated in terms of RMSNorm as follows:

$$\text{LayerNorm}(\mathbf{Z}) = \text{RMSNorm}(\mathbf{ZM}) \cdot \text{diag}(\boldsymbol{\alpha})\sqrt{D} + \mathbf{1}_N\boldsymbol{\beta}^\top,$$

where $\boldsymbol{\alpha}$ and $\boldsymbol{\beta}$ are learnable scale and offset parameters, respectively, specific to each LayerNorm instance. The matrix $\mathbf{M} = \mathbb{I}_D - \frac{1}{D}\mathbf{1}\mathbf{1}^\top$ centers each row of $\mathbf{Z}$ by subtracting its mean. The matrix $\mathbf{M}$ and $\text{diag}(\boldsymbol{\alpha})\sqrt{D}$ can then be absorbed in preceding and subsequent linear layers.

Moreover, since the orthogonal transformation $\mathbf{O}$ can be chosen as a rectangular matrix with orthonormal columns ($\mathbf{O} \in \mathbb{R}^{M \times N}$, $M \geq N$), this symmetry enables width-expanding transformations that preserve functionality. We provide a detailed derivation of this property in Appendix E, and demonstrate its empirical utility in Section 5.

Figure 2 illustrates how applying any orthogonal matrix $\mathbf{O}$ to the residual stream of a Transformer—after RMSNorm reparameterization—yields a functionally equivalent model.

# 4 Method

Building on the symmetry framework from Section 3.1, we now describe methods for aligning two independently trained Transformer models. The goal is to find function-preserving transformations - constrained to the relevant symmetry classes - that bring the models into structural and representational agreement.

Concretely, given a Transformer with $L$ layers, hidden dimension $d_h$, residual embedding size $d_r$, and $H$ attention heads per layer, alignment involves estimating a set of symmetry-constrained matrices: (i) a global orthogonal matrix $\mathbf{O} \in \mathbb{R}^{d_r \times d_r}$ for the residual stream; (ii) $L$ permutation matrices $\mathbf{P}_{FF} \in \mathbb{R}^{d_h \times d_h}$ for aligning neurons in the feed-forward layers; and (iii) $L$ semi-permutation matrices $\tilde{\mathbf{P}}_H \in \mathbb{R}^{H \times H}$ for aligning attention heads across layers.

We consider three strategies for estimating these transformations. *Activation matching* aligns layers by comparing intermediate activations on a shared dataset. *Weight matching* aligns parameters directly by minimizing distance under the symmetry constraints. *Learned matching* treats alignment as an optimization problem, learning the symmetry-aware re-parameterizations end-to-end. For activation matching, we use the method introduced by Verma and Elbayad [2024]. See Appendix C for an ablation study of our proposed algorithms.

## 4.1 Weight matching

We adapt the weight-based alignment strategy introduced in Ainsworth et al. [2022], which formulates alignment as an optimization problem over permutation matrices that maximize weight similarity across networks. The core intuition is that if two units across models have similar incoming and outgoing weights, they will likely implement similar functions and thus be aligned.

For Transformer feed-forward layers, we adopt a layerwise version of the "sum of bilinear assignments problem" (SOBLAP) proposed by Ainsworth et al. [2022]. Given weights $\mathbf{W}_\ell^{(A)}$ and $\mathbf{W}_\ell^{(B)}$ in layer $\ell$, we search for a permutation $\mathbf{P}_\ell^{\text{FF}}$ that maximizes alignment:

$$\mathbf{P}_\ell^{\text{FF}} = \arg\max_{\mathbf{P} \in \mathcal{S}_d} \langle \mathbf{W}_\ell^{(A)}, \mathbf{P}\mathbf{W}_\ell^{(B)}\mathbf{O}^\top \rangle_F + \langle \mathbf{W}_{\ell+1}^{(A)}, \mathbf{O}\mathbf{W}_{\ell+1}^{(B)}\mathbf{P}^\top \rangle_F, \tag{3}$$

where $\mathcal{S}_d$ is the set of permutation matrices, and $\mathbf{O}$ is the orthogonal matrix for the residual stream. This objective is NP-hard, but can be approximated using a coordinate descent strategy where each $\mathbf{P}_\ell^{\text{FF}}$ is updated by solving a linear assignment problem conditioned on adjacent layers.

For attention layers, we exploit the fact that QK and OV circuits are invariant under invertible reparameterizations [Elhage et al., 2021]. We define the **QK circuit** and **OV circuit** for each head $i$ as:

$$\text{QK}_i := (\mathbf{O}^\top \mathbf{W}_i^Q)(\mathbf{O}^\top \mathbf{W}_i^K)^\top, \quad \text{OV}_i := \mathbf{O}^\top \mathbf{W}_i^V \mathbf{W}_i^O \mathbf{O}.$$

We then define a cost matrix for aligning heads between models A and B using the Frobenius norm:

$$C_{i,j}^{\text{head}} = \|\text{QK}_i^{(A)} - \text{QK}_j^{(B)}\|_F^2 + \|\text{OV}_i^{(A)} - \text{OV}_j^{(B)}\|_F^2.$$

We solve a linear assignment problem to obtain the head-level permutation matrix $\mathbf{P}_\ell^{\text{H}}$.

For the residual stream, we estimate one global orthogonal matrix $\mathbf{O}$ by solving the Procrustes problem:

$$\mathbf{O} = \arg\min_{\mathbf{O} \in \mathbb{R}^{d_r \times d_r}, \mathbf{O}^\top \mathbf{O} = \mathbf{I}} \|\mathbf{R}^{(A)} - \mathbf{R}^{(B)}\mathbf{O}\|_F^2, \tag{4}$$

where $\mathbf{R}^{(A)}$ and $\mathbf{R}^{(B)}$ are the weights along the residual path collected from both models. The closed-form solution is given by the SVD of $\mathbf{R}^{(B)\top}\mathbf{R}^{(A)}$.

## 4.2 Learned matching

While activation and weight matching rely on static alignment criteria, learned matching directly optimizes the alignment parameters using task loss as supervision. Rather than aligning weights or activations explicitly, we treat the symmetry transformations themselves as trainable parameters, to be learned end-to-end (see Algorithm 1).

---

**Algorithm 1** Learning matching via task loss

---

**Require:** Base models $\boldsymbol{\theta}_A, \boldsymbol{\theta}_B$; Dataset $\mathcal{D}$; Iterations $N_{\text{iter}}$; Adam optimizer ($\texttt{lr} = \eta$).

1: Initialize latent matrices $\boldsymbol{Z}_{\text{FF}}, \boldsymbol{Z}_{\text{H}}, \boldsymbol{Z}_{\text{O}}$ using weight matching.

2: **for** $t = 1$ **to** $N_{\text{iter}}$ **do**

3:      Project: $\{\boldsymbol{P}_{\text{FF}}, \boldsymbol{P}_{\text{H}}, \boldsymbol{O}\} \leftarrow \{\text{PROJPERM}(\boldsymbol{Z}_{\text{FF}}), \text{PROJPERM}(\boldsymbol{Z}_{\text{H}}), \text{PROJORTH}(\boldsymbol{Z}_{\text{O}})\}$.

4:      Align: $\boldsymbol{\theta}_B^{\text{aligned}} \leftarrow \pi(\boldsymbol{\theta}_B; \boldsymbol{P}_{\text{FF}}, \boldsymbol{P}_{\text{H}}, \boldsymbol{O})$.                $\triangleright$ $\pi$ applies transformations

5:      Sample: $\lambda \sim \mathcal{U}(0.4, 0.6)$

6:      Sample batch $B = \{(\boldsymbol{X}_i, \boldsymbol{Y}_i)\}_{i=1}^{|B|}$ from $\mathcal{D}$.

7:      Interpolate: $\boldsymbol{\theta}_{\text{INTERP}} \leftarrow \lambda \cdot \boldsymbol{\theta}_A + (1 - \lambda) \cdot \boldsymbol{\theta}_B^{\text{aligned}}$.

8:      Objective: $\mathcal{J} \leftarrow \frac{1}{|B|} \sum_{(\boldsymbol{X}, \boldsymbol{Y}) \in B} \mathcal{L}_{\text{CE}}(\boldsymbol{\theta}_{\text{INTERP}}; \boldsymbol{X}, \boldsymbol{Y})$.

9:      Gradients: $(\boldsymbol{g}_{\boldsymbol{Z}_{\text{FF}}}, \boldsymbol{g}_{\boldsymbol{Z}_{\text{H}}}, \boldsymbol{g}_{\boldsymbol{Z}_{\text{O}}}) \leftarrow \nabla_{(\boldsymbol{Z}_{\text{FF}}, \boldsymbol{Z}_{\text{H}}, \boldsymbol{Z}_{\text{O}})} \mathcal{J}$.         $\triangleright$ STE for $\boldsymbol{Z}_{\text{FF}}, \boldsymbol{Z}_{\text{H}}$

10:     Update: For $k \in \{\text{FF,H,O}\}$, $\boldsymbol{Z}_k \leftarrow \text{Adam}(\boldsymbol{Z}_k, \boldsymbol{g}_{\boldsymbol{Z}_k}, \eta)$.

11: **end for**

12: Final projections: $\boldsymbol{P}_{\text{FF}}^* \leftarrow \text{PROJPERM}(\boldsymbol{Z}_{\text{FF}}), \boldsymbol{P}_{\text{H}}^* \leftarrow \text{PROJPERM}(\boldsymbol{Z}_{\text{H}}), \boldsymbol{O}^* \leftarrow \text{PROJORTH}(\boldsymbol{Z}_{\text{O}})$.

13: **return** $\boldsymbol{P}_{\text{FF}}^*, \boldsymbol{P}_{\text{H}}^*, \boldsymbol{O}^*$.

---

We introduce unconstrained latent matrices $\mathbf{Z}_{\text{FF}}, \mathbf{Z}_{\text{H}}$, and $\mathbf{Z}_{\text{O}}$, which are projected to the respective symmetry classes at each forward pass:

$$\mathbf{P}_{\text{FF}} = \text{PROJPERM}(\mathbf{Z}_{\text{FF}}), \quad \mathbf{P}_{\text{H}} = \text{PROJPERM}(\mathbf{Z}_H), \quad \mathbf{O} = \text{PROJORTH}(\mathbf{Z}_O). \tag{5}$$

We initialize these latent matrices using the weight matching procedure described in Section 4.1. For the permutation matrices, PROJPERM projects each matrix to the nearest permutation via the Hungarian algorithm; we use a straight-through estimator to allow gradients to flow through the relaxed $\mathbf{Z}$ parameters. The orthogonal projection PROJORTH computes $\mathbf{U}\mathbf{V}^\top$ from the SVD of $\mathbf{Z}_O = \mathbf{U}\boldsymbol{\Sigma}\mathbf{V}^\top$. This operation is fully differentiable.

At each step, we interpolate between model $A$ and the reparameterized model $B$ using a uniformly randomly sampled interpolation coefficient $\lambda \sim \mathcal{U}(0.4, 0.6)$:

$$\boldsymbol{\theta}_{\text{INTERP}} = \lambda \cdot \boldsymbol{\theta}_A + (1 - \lambda) \cdot \pi(\boldsymbol{\theta}_B),$$

where $\pi(\cdot)$ denotes alignment of model $B$. We then compute the original training loss on $\boldsymbol{\theta}_{\text{INTERP}}$ and backpropagate through the alignment transformation.

This approach encourages Transformer similarity through task performance, rather than explicit similarity of parameters or activation vectors, thus enabling the joint optimization of all symmetry-aware transformations over all network layers.

## 4.3 Multi-model merging

**Universe matching.** Following Crisostomi et al. [2024], we build a shared *universe* $U^{(t)}$ that serves as a common reference for all models. Given trained models $\boldsymbol{\theta}_1, \ldots, \boldsymbol{\theta}_M$, initialize $U^{(0)} \leftarrow \boldsymbol{\theta}_s$ with any seed model ($s \in \{1, \ldots, M\}$). For each iteration $t = 1{:}N$,

$$\pi_m^{(t)} \leftarrow \text{ALIGN}(\boldsymbol{\theta}_m, U^{(t-1)}) \quad \forall m, \qquad U^{(t)} \leftarrow \frac{1}{M} \sum_{m=1}^{M} \pi_m^{(t)}(\boldsymbol{\theta}_m).$$

Table 2: Loss barrier (lower is better) for each alignment method. Reported values are mean ± standard error, with rank in parentheses. Models are width-homogeneous, and only two models are aligned with each other. Rows highlighted in color correspond to our methods, using the same colors as in subsequent figures; bold text indicates the best performance for each dataset.

| Method | ViT | | | GPT-2 | |
| --- | --- | --- | --- | --- | --- |
| | CIFAR-10 | CIFAR-100 | Tiny ImageNet | Tiny Shakespeare | BookCorpus |
| Vanilla averaging | $1.69 \pm 0.07$ (5) | $2.46 \pm 0.04$ (5) | $2.84 \pm 0.02$ (5) | $2.02 \pm 0.12$ (5) | $4.34 \pm 0.09$ (5) |
| Activation matching | $1.27 \pm 0.13$ (4) | $2.11 \pm 0.17$ (4) | $1.86 \pm 0.10$ (4) | $1.43 \pm 0.16$ (4) | $4.05 \pm 0.13$ (4) |
| Weight matching (ours) | $0.36 \pm 0.01$ (2) | $0.69 \pm 0.21$ (3) | $0.47 \pm 0.04$ (3) | $0.34 \pm 0.01$ (2) | $1.56 \pm 0.02$ (2) |
| Learned matching (permutations) | $0.45 \pm 0.02$ (3) | $0.53 \pm 0.07$ (2) | $0.29 \pm 0.02$ (2) | $0.63 \pm 0.17$ (3) | $1.60 \pm 0.04$ (3) |
| **Learned matching (ours)** | $\mathbf{0.00 \pm 0.00}$ (1) | $\mathbf{0.00 \pm 0.00}$ (1) | $\mathbf{0.00 \pm 0.00}$ (1) | $\mathbf{0.02 \pm 0.00}$ (1) | $\mathbf{0.42 \pm 0.01}$ (1) |

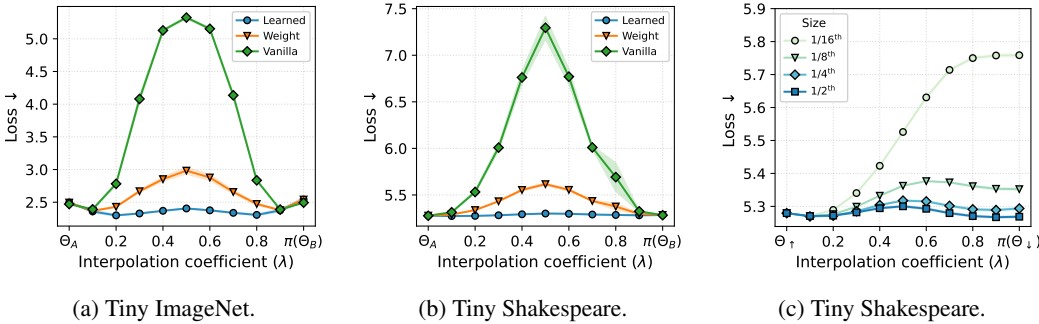

(a) Tiny ImageNet.    (b) Tiny Shakespeare.    (c) Tiny Shakespeare.

Figure 3: Loss along the interpolation path for (a, b) width-homogeneous and (c) width-heterogeneous alignment. In (c) $\Theta_\downarrow$ denotes the smaller model (reduced embedding dimension; the reduction ratio is indicated in the label), which is aligned to the larger one ($\Theta_\uparrow$, with embedding dimension 512). Learned matching was used for the width-heterogeneous results.

Each ALIGN step estimates a function-preserving transformation $\pi_m^{(t)}$ constrained to the symmetry classes of Section 3.1. The evolving anchor $U^{(t)}$ aggregates aligned parameters into a unified coordinate system, and after $N$ iterations the resulting $\{\pi_m^{(N)}\}$ approximately minimize the multi-model barrier in Eq. (2).

**Learned refinement.** We refine $\{\pi_m^{(N)}\}$ using the learned matching method from Section 4.2, extended to $M$-way mixtures. Sampling $\boldsymbol{\lambda} \sim \mathrm{Dirichlet}(\alpha \mathbf{1}_M)$ with $\alpha = 0.1$, we minimize

$$\mathcal{J} = \mathbb{E}_{\boldsymbol{\lambda}}\left[ \mathcal{L}\Big[ \sum_{m=1}^{M} \lambda_m \, \pi_m(\boldsymbol{\theta}_m)\Big](\mathcal{D}) - \sum_{m=1}^{M} \lambda_m \, \mathcal{L}[\pi_m(\boldsymbol{\theta}_m)](\mathcal{D})\right],$$

which directly drives $\mathcal{B}[\{\pi_m(\boldsymbol{\theta}_m)\}](\mathcal{D})$ toward zero. Gradients are backpropagated through $\pi_m$ via the projection operators of Section 4.2.

## 5  Results

We evaluate the proposed model alignment methods on two Transformer architectures: Vision Transformers (ViTs) and GPT-2, spanning vision and language tasks. To measure LMC, we compute the loss barrier between two models $\boldsymbol{\theta}_A$ and $\boldsymbol{\theta}_B$ as defined in Equation 1 on the test split. A lower barrier indicates better connectivity; the optimal value is $\mathcal{B} = 0$. See Appendix C and D for further results.

**Two-way model alignment.** Table 2 and Figure 3 summarize alignment between two independently trained models. Our learned matching method consistently outperforms all alternative approaches, achieving zero or near-zero barriers on every dataset except BookCorpus. The weight-matching variant—fully unsupervised and training-free—also yields substantial reductions and often surpasses permutation-only learned matching (which is akin to the STE-based approach in

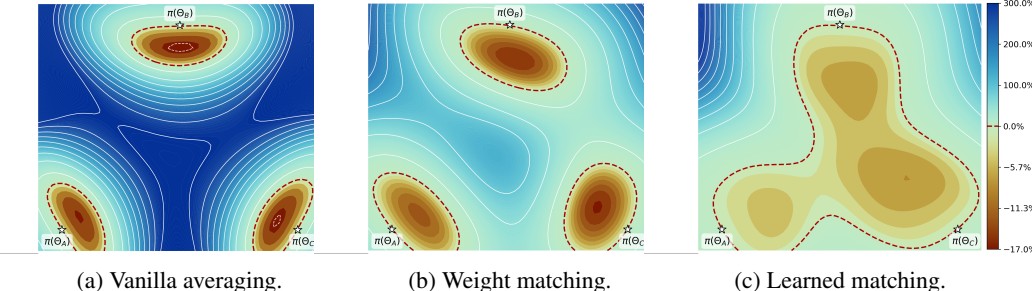

(a) Vanilla averaging.    (b) Weight matching.    (c) Learned matching.

Figure 4: Linear mode connectivity surface between three CIFAR-10 models for different alignment strategies. Colors show the relative loss deviation from the linear interpolation baseline across the simplex spanned by $\pi(\Theta_A)$, $\pi(\Theta_B)$, and $\pi(\Theta_C)$. The dashed contour ($\varepsilon = 0$) marks points where the loss equals the linear baseline.

Git Rebasin). The degree of connectivity among Transformer minima is striking: for comparison, achieving a zero barrier for ResNet-20 on CIFAR-10 in Git Rebasin required a $32\times$ width increase [Ainsworth et al., 2022]. Figure 3c further considers width-heterogeneous alignment, aligning a larger model with embedding dimension 512 to smaller counterparts. Despite the architectural mismatch, the interpolation paths remain at or near zero barrier, indicating connected regions across Transformer sizes.

While image tasks reliably attain $\mathcal{B} \approx 0$—including Tiny ImageNet—the language experiments, particularly on BookCorpus, exhibit higher barriers. This may reflect imperfect alignment, or genuinely disconnected minima. Juneja et al. [2023] show that, for NLP classifiers, fine-tuning can yield multiple basins associated with distinct generalization strategies (e.g., lexical-overlap vs. syntactic cues). Thus, while models using the same strategy might be linearly connected, linear paths across strategies exhibit barriers. This suggests the non-zero barriers on BookCorpus may reflect fundamentally different solutions rather than merely suboptimal alignment.

**Multi-way model alignment.** Figure 4 extends the analysis to aligning three independently trained CIFAR-10 models. We visualize the loss over the simplex spanned by $\pi(\Theta_A)$, $\pi(\Theta_B)$, and $\pi(\Theta_C)$. Relative to vanilla averaging, both weight matching and learned matching flatten the surface toward the linear-interpolation baseline, with learned matching producing the broadest region of near-zero deviation. These results suggest that our procedures connect not only pairs of solutions but also carve out a shared low-loss manifold spanning multiple models.

## 6   Understanding the gap between weight- and learned matching

Weight matching is an attractive, data-free alignment method: it operates directly on parameters, requires no training data, and is computationally efficient—making it practical for settings such as federated learning where data sharing is limited. However, it typically yields higher interpolation barriers than learned matching, which optimizes alignment with task-loss supervision. This raises a key question: *where does the gap arise, and can parameter-only methods close it?*

To locate the gap, we analyze the learned transformations. Across runs, the permutations of attention heads and feed-forward blocks found by weight matching remain essentially unchanged under learned matching. The difference concentrates in the orthogonal map **O** that aligns the residual stream. As shown in Fig. 5, the eigen-angles of $\mathbf{O}_{\mathrm{WM}}$ are broadly distributed over $[0, 2\pi]$, indicating nearly arbitrary rotations/reflections, whereas the relative correction $\mathbf{O}_{\mathrm{diff}} = \mathbf{O}_{\mathrm{LM}}\mathbf{O}_{\mathrm{WM}}^{\top}$ concentrates near 0 mod $2\pi$, i.e., learned matching makes small, targeted refinements.

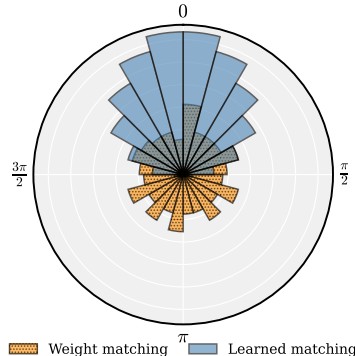

Figure 5: Eigenvalue spectra of $\mathbf{O}_{\mathrm{WM}}$ and $\mathbf{O}_{\mathrm{diff}} = \mathbf{O}_{\mathrm{LM}}\mathbf{O}_{\mathrm{WM}}^{T}$. The latter indicates small refinements from learned matching.

In short, learned matching primarily *refines* the orthogonal alignment provided by weight matching. This suggests a promising avenue: better, data-free estimation of $\mathbf{O}$ (e.g., with structural priors or spectral objectives) could narrow most of the performance gap without full supervision.

## 7 Related work

**Linear Mode Connectivity (LMC).** LMC describes the existence of low-loss linear paths between independently trained networks. Early work on mode connectivity [Garipov et al., 2018, Draxler et al., 2018] identified nearly constant-loss non-linear paths, suggesting SGD solutions lie on a connected manifold. LMC focuses on linear interpolations: Entezari et al. [2021] conjectured that permutation symmetries account for the observed disconnection, and once resolved, SGD solutions lie in a single basin. While a formal proof remains open, empirical evidence increasingly supports this view (see below). Recent work also links global LMC to layer-wise linearity [Zhou et al., 2024, Adilova et al., 2023].

**Model merging and symmetry alignment.** Several approaches have exploited the symmetry structure of neural networks to enable one-shot model merging. OT fusion [Singh and Jaggi, 2020] casts neuron alignment as an optimal transport (OT) problem, computing the Wasserstein barycenter between corresponding layers based on activation or weight similarity. This enables data-driven model fusion that outperforms naive averaging and can approximate ensemble performance after moderate finetuning. Related methods include Liu et al. [2022], Akash et al. [2022]. Subsequently, Git Re-Basin [Ainsworth et al., 2022] considers three alignment methods (two of which are highly similar to prior work of [Singh and Jaggi, 2020]): activation matching, weight matching (WM), and a learning-based variant using straight-through estimators (STE) [Bengio et al., 2013]. WM is data-free but loss landscape-agnostic; STE backpropagates through soft permutations and depends critically on WM for initialization. This approach achieves zero-barrier interpolation for modified (widened) ResNets with LayerNorm, a result later extended to BatchNorm networks via statistical recalibration [Jordan et al., 2022].

Ito et al. [2025] show that WM aligns dominant singular directions without altering singular values, thereby enabling LMC while preserving functionality. Consequently, it exhibits higher transitivity than STE, which may overfit local loss geometry. Sinkhorn Re-Basin [Peña et al., 2022] instead optimizes relaxed permutations using Sinkhorn operator and implicit differentiation [Eisenberger et al., 2022] to reduce interpolation barriers.

Transformer-specific extensions have also emerged: Imfeld et al. [2023] adapts OT fusion to handle multi-head attention and residual structures using Sinkhorn-based soft alignment, while Verma and Elbayad [2024] uses correlation-based matching to align BERT models. Both show reduced loss barriers compared to naive averaging, though non-zero barriers remain.

Beyond merging identical-task models, Stoica et al. [2024] tackles multi-task model merging by aligning both inter- and intra-model features, revealing redundancy in neural representations. Cycle-consistent alignment across multiple models is explored in Crisostomi et al. [2024], enforcing consistency of neuron permutations to support multi-way merging.

## 8 Conclusion

We introduced a unified framework for symmetry-aware model alignment that captures a broad class of transformations—permutation, semi-permutation, orthogonal, and general invertible maps. This generalization subsumes prior re-basin techniques and enables, for the first time, the discovery of low- and zero-loss linear interpolation paths between independently trained Vision Transformers and GPT-2 models. Our empirical results show that broader symmetry classes are essential to uncovering the connectedness of modern neural network loss landscapes. These findings highlight the importance of modeling and leveraging richer symmetries in reparameterization to advance our understanding of neural network geometry, with potential implications for model ensembling, transfer, and interoperability. A more extensive discussion of the theoretical implications and broader impact of these results is provided in Appendix F, and the practical limitations of our framework are summarized in Appendix A.

## Acknowledgements

Alexander Theus and Sidak Pal Singh acknowledge the financial support from the Max Planck ETH Center for Learning Systems. Antonio Orvieto acknowledges the financial support of the Hector Foundation.

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

Table 3: Configuration and training details for Vision Transformer (ViT) and GPT-2 across two datasets each.

| Component | ViT | | GPT-2 | |
|---|---|---|---|---|
| | **CIFAR-10/100** | **Tiny ImageNet** | **Tiny Shakespeare** | **BookCorpus** |
| Transformer layers | 6 | 8 | 6 | 6 |
| Attention heads | 8 | 8 | 4 | 8 |
| Embedding dimension | 256 | 384 | 256 | 512 |
| MLP hidden dimension | 512 | 768 | 1024 | 2048 |
| Patch size | $4 \times 4$ | $8 \times 8$ | – | – |
| Sequence length | – | – | 256 | 512 |
| Training epochs | 150 | 150 | 100 | 5 |
| Batch size | 128 | 128 | 32 | 64 |
| Optimizer | AdamW | AdamW | AdamW | AdamW |
| Learning rate | $3 \times 10^{-4}$ | $3 \times 10^{-4}$ | $3 \times 10^{-4}$ | $2.5 \times 10^{-4}$ |
| Weight decay | $1 \times 10^{-3}$ | 0.05 | 0.01 | 0.01 |
| Learning rate schedule | Cosine annealing | Cosine (warmup) | Cosine (warmup) | Cosine (warmup) |
| Hardware | 1× RTX 2060 | 1x RTX 4090 | 1× RTX 4090 | 4× RTX 4090 |

# A  Limitations

While the research introduces a unified framework for symmetry-aware model alignment in Transformers, it presents some limitations and areas for future exploration. Our methodology introduces a generalized notion of LMC that, in some cases, requires reparameterization (e.g., RMSNorm reparameterization or multiplying out intra-head dependencies). Although it preserves functional equivalence, it alters the underlying network architecture. The empirical results for language models were based on a smaller version of GPT-2 language models with reduced parameters due to resource constraints (Appendix B), indicating the need for evaluation on larger, more contemporary language models. The current scope is focused on aligning pairs of models that have been pretrained on the same task, and further investigation could extend these methodologies to models trained on (partially) different tasks [Stoica et al., 2024]. Additionally, while additional results (Appendix D) demonstrate that soft permutations can improve test-time performance of the merged model, more work is needed to fully refine soft relaxations of symmetry operations to improve the performance of aligned models. Finally, due to computational constraints, the study utilizes standard benchmarks such as CIFAR-10/100 and Tiny ImageNet for Vision Transformers and TinyShakespeare and Book-Corpus for GPT-2; exploring performance on a broader or more complex range of benchmarks could further validate the findings.

# B  Experimental details

We provide implementation and training details for the Vision Transformer (ViT) and GPT-2 models used in our experiments. Table 3 summarizes the key architectural parameters, optimization settings, and hardware configurations for both models. Additional details on data preprocessing, augmentation, and training protocols are provided in the following subsections.

### B.1  Vision Transformer (ViT)

We trained two Vision Transformer (ViT) configurations, one for CIFAR-10/100 and another for Tiny ImageNet. For CIFAR-10/100, the model consisted of 6 transformer layers and 8 attention heads, with an embedding dimension of 256 and a feedforward (MLP) hidden dimension of 512. Input images were divided into non-overlapping patches of size $4 \times 4$.

For Tiny ImageNet, the model was scaled up to 8 transformer layers with the same number of attention heads (8). The embedding and hidden dimensions were increased to 384 and 768, respectively, and the patch size was enlarged to $8 \times 8$ to accommodate higher-resolution inputs.

All ViT models were trained for 150 epochs using the AdamW optimizer with an initial learning rate of $3 \times 10^{-4}$. For CIFAR-10/100, we applied a weight decay of $1 \times 10^{-3}$, while for Tiny ImageNet

we used $0.05$. Both configurations used cosine learning rate scheduling; the Tiny ImageNet setup additionally employed a short warmup phase at the start of training.

For CIFAR-10/100, standard data augmentation was applied, including random cropping with padding (crop size 32, padding 4), horizontal flipping, and color jittering (brightness, contrast, and saturation adjustments of 0.4, and hue variation of 0.1). For Tiny ImageNet, a more advanced augmentation pipeline was used, consisting of random resized cropping to $64 \times 64$ (scale range $(0.8, 1.0)$, aspect ratio $(0.9, 1.1)$), random horizontal flipping with $p = 0.5$, and the `AutoAugment` policy for ImageNet. All images were normalized using the dataset-specific mean and standard deviation.

Additionally, for Tiny ImageNet, we applied post-hoc temperature scaling to rescale the model logits and improve confidence calibration.

Training for the CIFAR-10/100 model was performed on a single NVIDIA GeForce RTX 2060 GPU, while the Tiny ImageNet model was trained on a single NVIDIA RTX 4090 GPU. On the Tiny ImageNet test dataset, the models achieve an accuracy of $44.19 \pm 0.17$ and a calibrated loss of $2.54 \pm 0.02$. For the CIFAR-10 test dataset, they obtain an accuracy of $83.81 \pm 0.44$ and a loss of $0.57 \pm 0.01$.

## B.2  GPT-2

Two small-scale GPT-2 models were trained: one on the Tiny Shakespeare corpus and another on the BookCorpus dataset.

For Tiny Shakespeare, the model consisted of 6 transformer layers with 4 attention heads, an embedding dimension of 256, and an MLP hidden dimension of 1024. Sequences were truncated or padded to 256 tokens. Training was performed for 100 epochs with a batch size of 32 using the AdamW optimizer. The learning rate was set to $3 \times 10^{-4}$ with a cosine learning rate schedule and warmup phase, and a weight decay of 0.01. Additionally, early stopping was performed with a patience of 5 epochs to prevent overfitting. Training was conducted on a single NVIDIA RTX 4090 GPU. The model achieves a test loss of $5.28 \pm 0.00$.

For BookCorpus, the model used 6 transformer layers and 8 attention heads, with an embedding dimension of 512 and an MLP hidden dimension of 2048. Tokenized sequences were limited to 512 tokens using the GPT-2 tokenizer, with end-of-sequence tokens used for padding. The model was trained for 5 epochs using the AdamW optimizer with an initial learning rate of $2.5 \times 10^{-4}$, a 5% warmup ratio, and a weight decay of 0.01. Mixed-precision (fp16) training was enabled to improve throughput. This model was trained across four NVIDIA RTX 4090 GPUs with an effective batch size of 64 (16 per device). On this dataset, the models obtain a loss of $3.55 \pm 0.01$ on the test split.

## B.3  Merging

To merge the ViT and GPT-2 models, we use the same setup as for training the individual models.

# C  Ablation study

## C.1  Learned matching

### C.1.1  Coefficient sampling

In Section 5, we already ablated the effect of using learned permutations in place of orthogonal maps, highlighting the importance of capturing more general alignment symmetries. Here, we further investigate how the choice of interpolation coefficient sampling strategy influences performance (see Line 5 in Algorithm 1). Specifically, we compare four sampling schemes:

- **Fixed interpolation ($\lambda = 0.5$):** A deterministic strategy where $\lambda$ is always set to 0.5, representing a balanced average of the two models.

- **Uniform sampling [$0.4, 0.6$]:** A narrow uniform distribution centered at 0.5, introducing small random perturbations around equal weighting.

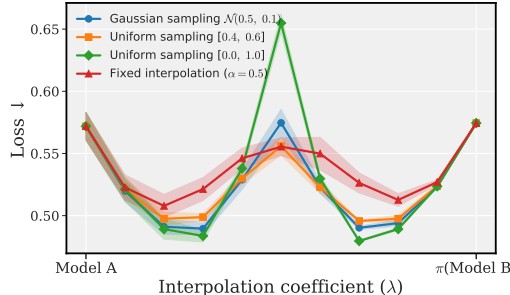

Figure 6: Loss curves for different strategies of sampling the interpolation coefficient $\lambda$. Each line shows the mean loss, with shaded areas denoting standard deviation. Experiments were carried out on ViTs trained on CIFAR-10.

Figure 7: Loss barriers over multiple iterations of weight matching, comparing orthogonal vs. permutation matching for the residual weights. Experiments were carried out on ViTs trained on CIFAR-10.

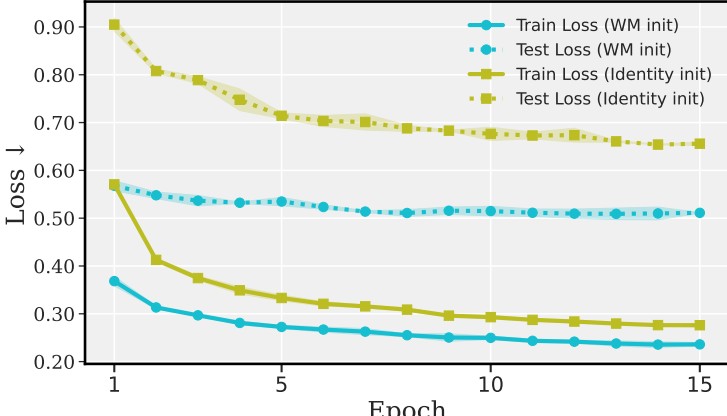

Figure 8: Comparison of train and test loss over epochs for learned matching, using weight matching ("WM init") versus identity initialization ("Identity init"). Using weight matching as an initialization point yields consistently lower loss. Model: ViT. Dataset: CIFAR-10.

- **Uniform sampling [**$0.0, 1.0$**]:** A broad uniform distribution over the entire interpolation range, allowing any convex combination of the two models.

- **Gaussian sampling** $\mathcal{N}(0.5, 0.1)$**:** A stochastic strategy that samples from a Gaussian centered at 0.5 with standard deviation 0.1, clipped to the interval $[0, 1]$.

We visualize the impact of these sampling strategies in Figure 6, where we report the mean and standard deviation of loss values along the interpolation path for ViTs trained on CIFAR-10. Notably, both uniform and Gaussian sampling result in relatively high loss barriers, indicating unstable interpolations, particularly around $\lambda = 0.5$. In contrast, narrow uniform sampling and fixed interpolation produce lower loss near the midpoint. However, fixed interpolation exhibits significantly higher loss in the surrounding regions, leading to elevated barriers for certain random seeds and greater variance overall. Based on these observations, we adopt narrow uniform sampling in our implementation, as it offers more consistent performance across different random seeds.

### C.1.2 Initialization

The results reported in Section 5 use *weight matching* to initialize the function-preserving permutation and orthogonal transformations. In this section, we evaluate the effectiveness of weight matching as an initialization strategy by comparing it to a baseline with no prior matching.

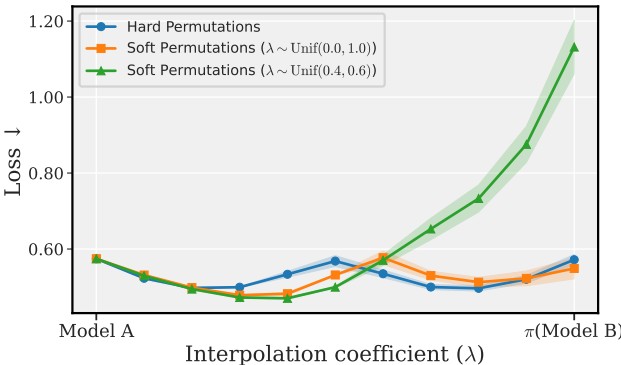

Figure 9: Loss barrier across the interpolation path using soft permutations. Plotted are the mean test loss and shaded regions representing $\pm 1$ standard deviation over 3 seeds. Model: ViT. Dataset: CIFAR-10.

We train the learned matching procedure for 15 epochs and report both training and test losses for ViTs trained on CIFAR-10 across four random seeds (see Figure 8). Initializing with weight matching leads to significantly lower training and test losses. Notably, models reach zero test loss barriers within just one to two epochs of training. In contrast, training without any prior matching results in consistently higher losses, with positive loss barriers remaining even after 15 epochs.

Evidently, learning the transformations alone is insufficient; a strong initialization provided by weight matching is essential for achieving low loss barriers and faster convergence.

## C.2 Weight matching

Our proposed iterative weight matching algorithm, while not outperforming learned matching, requires significantly fewer computational resources and has the added advantage of being completely data-free. Nevertheless, several questions remain. In particular: (1) Does orthogonal matching provide improvements over permutation matching, as observed in the learned variant (see Table 2)? (2) How many iterations are needed for convergence?

To address these questions, we evaluate the loss barrier across different numbers of iterations for both permutation and orthogonal matching. Results for ViTs trained on CIFAR-10 are shown in Figure 7. A stark contrast emerges: for orthogonal matching, the loss barrier steadily decreases, converging after five iterations with a substantially reduced barrier. In contrast, permutation-based matching shows no significant improvement across iterations.

These findings confirm that orthogonal matching provides a more effective path to convergence than permutation matching in the data-free setting, reinforcing its role in our proposed algorithm.

## D Soft-permutations

Soft permutations provide a continuous relaxation of hard (i.e., exact) permutations by allowing convex combinations of layer units. Formally, we define them as doubly stochastic matrices (i.e., matrices with non-negative entries whose rows and columns each sum to one) lying within the Birkhoff polytope, whose vertices correspond to the set of hard permutation matrices. This relaxation enables mappings that go beyond strict one-to-one neuron correspondences, offering greater flexibility in aligning network representations. We show results in Figure 9.

This section details the methodology for learning such soft permutation matrices to align the parameters of two models, $\boldsymbol{\theta}_A$ and $\boldsymbol{\theta}_B$. Unlike the hard permutations in Algorithm 1 which use a Straight-Through Estimator (STE), here soft permutations are derived from learnable latent parameters and made doubly stochastic using differentiable Sinkhorn normalization. The latter approach performed better than STE in our experiments.

## D.1 Objective

The primary goal is to learn a set of layer-wise latent matrices $\{\boldsymbol{Z}_l\}_l$. These latent matrices are transformed into soft permutation matrices $\{\boldsymbol{P}_l\}_l$ (where $\boldsymbol{P}_l = \text{Sinkhorn}(\exp(\boldsymbol{Z}_l))$) which are then used to construct a transformation $\pi$. This transformation aligns one model to another, e.g., yielding $\boldsymbol{\theta}_B^{\text{aligned}} = \pi(\boldsymbol{\theta}_B; \{\boldsymbol{P}_l\})$. The optimal latent matrices $\{\boldsymbol{Z}_l^*\}$ are those that minimize the empirical risk (loss) of an interpolated model over a batch $B$ drawn from the dataset $\mathcal{D}$:

$$\{\boldsymbol{Z}_l^*\} = \underset{\{\boldsymbol{Z}_l\}}{\arg\min} \frac{1}{|B|} \sum_{(\boldsymbol{X},\boldsymbol{Y})\in B} \left[\mathcal{L}_{\text{CE}}\left(\lambda \boldsymbol{\theta}_A + (1-\lambda)\,\pi(\boldsymbol{\theta}_B; \{\text{Sinkhorn}(\exp(\boldsymbol{Z}_l))\})\right)(\boldsymbol{X},\boldsymbol{Y})\right]$$

where $\lambda$ is an interpolation coefficient, typically sampled uniformly at random (e.g., $\lambda \sim \mathcal{U}[0.4, 0.6]$ as in Algorithm 1 or $\lambda \sim \mathcal{U}[0, 1]$). The optimization is performed with respect to latent parameters $\boldsymbol{Z}_l$.

## D.2 Parametrization and initialization of latent matrices

**Parametrization from latent matrices.** The soft permutation matrices $\boldsymbol{P}_l$ (which must be doubly stochastic) are not optimized directly. Instead, we optimize underlying unconstrained latent matrices $\boldsymbol{Z}_l$. To obtain $\boldsymbol{P}_l$ from $\boldsymbol{Z}_l$:

1. First, a non-negative matrix $\widetilde{\boldsymbol{P}}_l$ is generated using an element-wise exponential map:

$$\widetilde{\boldsymbol{P}}_l = \exp(\boldsymbol{Z}_l)$$

   This ensures all entries are positive, a requirement for the Sinkhorn algorithm, and allows unconstrained optimization of $\boldsymbol{Z}_l$ as gradients can flow back through the $\exp$ function.

2. Second, $\widetilde{\boldsymbol{P}}_l$ is projected onto the Birkhoff polytope $\mathbb{B}$ using the differentiable Sinkhorn-Knopp normalization (detailed below in the learning process) to yield the doubly stochastic soft permutation matrix $\boldsymbol{P}_l$.

Thus, $\boldsymbol{P}_l = \text{Sinkhorn}(\exp(\boldsymbol{Z}_l))$, and $\boldsymbol{Z}_l$ are the parameters learned via gradient descent.

**Initialization of latent matrices $\boldsymbol{Z}_l$.** The latent matrices $\boldsymbol{Z}_l$ are initialized by first constructing a target matrix $\boldsymbol{P}_l^0$, which represents a desired initial state for $\exp(\boldsymbol{Z}_l^0)$—before Sinkhorn normalization.

A baseline for random noise is established using a Xavier-scheme variance: let $\sigma^2 = \frac{2}{\texttt{fan-in}+\texttt{fan-out}}$. A scaling factor for noise is defined as $a = \varepsilon\,\sigma\sqrt{3}$, where $\varepsilon$ is a coefficient tuning the amount of noise to be injected. A random noise matrix $\boldsymbol{P}_l^{\text{rand}}$ is then sampled, with entries, for example, $[\boldsymbol{P}_l^{\text{rand}}]_{ij} \sim \text{Uniform}(0, 2a)$. Note one can add a small positive constant to $\boldsymbol{P}_l^0$ to ensure all entries are strictly positive.

The target matrix $\boldsymbol{P}_l^0$ is constructed based on one of the following strategies:

- **Random initialization:** The target matrix is formed directly from the random noise:

$$\boldsymbol{P}_l^0 = \boldsymbol{P}_l^{\text{rand}}$$

- **From pre-computed permutation:** If an initial hard permutation matrix $\boldsymbol{P}_l^{\text{init}}$ is available (e.g., from weight matching, as used for initializing $\boldsymbol{Z}_{\text{FF}}, \boldsymbol{Z}_{\text{H}}$ in Algorithm 1), the target matrix is formed by perturbing this known permutation:

$$\boldsymbol{P}_l^0 = \boldsymbol{P}_l^{\text{init}} + \boldsymbol{P}_l^{\text{rand}}$$

The initial latent parameters $\boldsymbol{Z}_l^0$ are then set by inverting the exponential parametrization, i.e., $\boldsymbol{Z}_l^0 = \log(\boldsymbol{P}_l^0)$, applied element-wise to the strictly positive target matrix $\boldsymbol{P}_l^0$. This ensures that $\exp(\boldsymbol{Z}_l^0) = \boldsymbol{P}_l^0$ at the start of the learning process.

## D.3 Learning process

The latent matrices $\boldsymbol{Z}_l$ for all relevant layers are optimized over $N_{\text{iter}}$ iterations. In each iteration $t$, using the Adam optimizer with learning rate $\eta$:

1. **Soft permutation matrix computation:**
   For each layer $l$:
   (a) Obtain the non-negative matrix from the current latent parameters: $\widetilde{\boldsymbol{P}}_l = \exp(\boldsymbol{Z}_l)$.
   (b) Project $\widetilde{\boldsymbol{P}}_l$ onto the Birkhoff polytope using $K$ iterations of the Sinkhorn-Knopp algorithm to get the soft permutation matrix $\boldsymbol{P}_l$. Let $\boldsymbol{Q}_l^{(0)} = \widetilde{\boldsymbol{P}}_l$. For $k = 1, \ldots, K$:

$$\boldsymbol{Q}_l^{(k)} \leftarrow \operatorname{diag}\left(\frac{1}{\boldsymbol{Q}_l^{(k-1)}\boldsymbol{1}}\right)\boldsymbol{Q}_l^{(k-1)} \qquad \text{(Normalize rows)}$$

$$\boldsymbol{Q}_l^{(k)} \leftarrow \boldsymbol{Q}_l^{(k)}\operatorname{diag}\left(\frac{1}{\boldsymbol{1}^\top\boldsymbol{Q}_l^{(k)}}\right) \qquad \text{(Normalize columns)}$$

   The resulting soft permutation is $\boldsymbol{P}_l = \boldsymbol{Q}_l^{(K)}$. This Sinkhorn normalization process is differentiable with respect to $\widetilde{\boldsymbol{P}}_l$.

2. **Model alignment and interpolation:**
   Align model $\boldsymbol{\theta}_B$ using the computed soft permutations $\{\boldsymbol{P}_l\}_l$: $\boldsymbol{\theta}_B^{\text{aligned}} \leftarrow \pi(\boldsymbol{\theta}_B; \{\boldsymbol{P}_l\}_l)$. Sample an interpolation coefficient $\lambda$ (e.g., $\lambda \sim \mathcal{U}[0.4, 0.6]$ as in Algorithm 1, or from $\mathcal{U}[0, 1]$ based on desired outcomes, see discussion below). Form the interpolated model: $\boldsymbol{\theta}_{\text{INTERP}} \leftarrow \lambda\boldsymbol{\theta}_A + (1 - \lambda)\boldsymbol{\theta}_B^{\text{aligned}}$.

3. **Loss computation and parameter update:**
   Compute the empirical cross-entropy loss $\mathcal{J}$ on a sampled batch $B = \{(\boldsymbol{X}_i, \boldsymbol{Y}_i)\}_{i=1}^{|B|}$ from dataset $\mathcal{D}$:

$$\mathcal{J} \leftarrow \frac{1}{|B|}\sum_{(\boldsymbol{X},\boldsymbol{Y})\in B}\mathcal{L}_{\text{CE}}(\boldsymbol{\theta}_{\text{INTERP}}; \boldsymbol{X}, \boldsymbol{Y})$$

   Compute gradients of the loss with respect to the latent parameters: $(\ldots, \nabla_{\boldsymbol{Z}_l}\mathcal{J}, \ldots)$. Update each latent matrix $\boldsymbol{Z}_l$ using the Adam optimizer:

$$\boldsymbol{Z}_l \leftarrow \operatorname{Adam}(\boldsymbol{Z}_l, \nabla_{\boldsymbol{Z}_l}\mathcal{J}, \eta)$$

   Since the exponential mapping and Sinkhorn normalization are differentiable, gradients flow directly back to the latent parameters $\boldsymbol{Z}_l$.

After $N_{\text{iter}}$ training iterations, the final optimized latent matrices $\{\boldsymbol{Z}_l^*\}$ are used to yield the learned soft permutation matrices $\{\boldsymbol{P}_l^* = \operatorname{Sinkhorn}(\exp(\boldsymbol{Z}_l^*))\}$.

## D.4 Remarks.

In our explorations, these learned soft permutations are applied to align components within Transformer architectures, specifically targeting attention heads and MLP layers between independently trained models. Empirically, this approach often yields meaningful improvements in the test-time performance of the merged (interpolated) model, particularly at the midpoint of the interpolation path ($\lambda \approx 0.5$). However, a key issue of using soft permutations is that exact functional equivalence at the aligned endpoint (i.e., for $\boldsymbol{\theta}_B^{\text{aligned}}$ compared to the original $\boldsymbol{\theta}_B$) is generally not maintained. We observe that the degree of functional equivalence at the endpoint can be improved by modifying the sampling strategy for the interpolation coefficient $\lambda$ during the learning process—for instance, by sampling $\lambda$ uniformly from the entire $[0, 1]$ range, which allows for more direct optimization of the transformation of the aligned endpoint. Nevertheless, this adjustment typically involves a trade-off: while endpoint functional equivalence may improve, the performance gain observed at the midpoint of the interpolation path might be reduced compared to when $\lambda$ is sampled more narrowly (e.g., from $\mathcal{U}[0.4, 0.6]$). We leave further exploration of this approach to future work.

# E    General symmetries of network components

Consider a feed-forward network with $L$ layers, where the $l$-th layer computes activation vector $a_l = \sigma_l(\boldsymbol{W}_l a_{l-1})$, with $a_0 = \boldsymbol{X}$ being the input. The full parameter set is $\boldsymbol{\theta} = \{\boldsymbol{W}_l\}_{l=1}^L$. Such a network implements the following composed mapping:

$$\boldsymbol{Y} = \boldsymbol{W}_L \circ \sigma_L \circ \boldsymbol{W}_{L-1} \circ \ldots \circ \boldsymbol{W}_1 \boldsymbol{X}$$

with:

$$\boldsymbol{\theta} = \begin{pmatrix} \boldsymbol{W}_1 & 0 & \cdots & 0 \\ 0 & \boldsymbol{W}_2 & \ddots & \vdots \\ \vdots & \ddots & \ddots & 0 \\ 0 & \cdots & 0 & \boldsymbol{W}_L \end{pmatrix}$$

We define transformation $\pi$ as the *alignment* that reparametrizes the network to account for its inherent symmetries, mapping the original parameters $\boldsymbol{\theta}$ to an equivalent (or approximately equivalent) set $\boldsymbol{\theta}' = \{\boldsymbol{W}_l'\}_{l=1}^L$, with the goal of achieving *linear mode connectivity*—i.e. maintaining low loss barrier along the interpolation path.

The reparameterization imposed by $\pi$ is typically achieved in practice by defining a set of (approximately) invertible matrices $\{\boldsymbol{S}_l\}_{l=0}^L$, where $\boldsymbol{S}_l \in \mathbb{R}^{d_l \times d_l}$ acts as a change of basis for the $d_l$-dimensional hidden representation $a_l$. We fix the input and output bases by setting $\boldsymbol{S}_0 = \boldsymbol{I}_{d_0}$ and $\boldsymbol{S}_L = \boldsymbol{I}_{d_L}$. The transformed weights are then given by:

$$\boldsymbol{W}_l' = \boldsymbol{S}_l \boldsymbol{W}_l \boldsymbol{S}_{l-1}^{-1} \quad \text{for } l \in \{1, \ldots, L\}.$$

The aligned network function becomes:

$$f[\boldsymbol{\theta}'](\boldsymbol{X}) = \sigma_L(\boldsymbol{W}_L \boldsymbol{S}_{L-1}^{-1} \sigma_{L-1}(\boldsymbol{S}_{L-1} \boldsymbol{W}_{L-1} \boldsymbol{S}_{L-2}^{-1} \ldots \sigma_1(\boldsymbol{S}_1 \boldsymbol{W}_1 \boldsymbol{X}) \ldots))$$

Exact functional invariance, $f[\boldsymbol{\theta}'](\boldsymbol{X}) = f[\boldsymbol{\theta}](\boldsymbol{X})$, is guaranteed if $\boldsymbol{S}_L = \boldsymbol{I}$ and the activation functions $\sigma_l$ are equivariant with respect to their corresponding transformations $\boldsymbol{S}_l$, i.e., $\boldsymbol{S}_l \sigma_l(Z) = \sigma_l(\boldsymbol{S}_l Z)$ for $l \in \{1, \ldots, L-1\}$.

A common case ensuring such equivariance is when $\sigma_l$ is an element-wise activation function (e.g., RELU—as seen in Section 3.1)—and $\boldsymbol{S}_l$ is a *permutation matrix* $\boldsymbol{P}_l$. In this scenario, the set of original parameters $\boldsymbol{\theta}$ can be represented as a block diagonal matrix $\boldsymbol{\theta}_{\text{diag}} = \text{diag}(\boldsymbol{W}_1, \boldsymbol{W}_2, \ldots, \boldsymbol{W}_L)$. The transformation $\boldsymbol{\theta}_{\text{diag}}' = \pi(\boldsymbol{\theta}_{\text{diag}})$ with blocks $\boldsymbol{W}_l' = \boldsymbol{P}_l \boldsymbol{W}_l \boldsymbol{P}_{l-1}^T$ can be expressed by defining block diagonal transformation matrices:

$$\boldsymbol{P}_{\text{left}} = \begin{pmatrix} \boldsymbol{P}_1 & 0 & \cdots & 0 \\ 0 & \boldsymbol{P}_2 & \ddots & \vdots \\ \vdots & \ddots & \ddots & 0 \\ 0 & \cdots & 0 & \boldsymbol{I}_L \end{pmatrix}, \qquad \boldsymbol{P}_{\text{right}} = \begin{pmatrix} \boldsymbol{I}_0 & 0 & \cdots & 0 \\ 0 & \boldsymbol{P}_1^\top & \ddots & \vdots \\ \vdots & \ddots & \ddots & 0 \\ 0 & \cdots & 0 & \boldsymbol{P}_{L-1}^\top \end{pmatrix}$$

Then, the transformed parameters are obtained by block-wise operations: $(\boldsymbol{\theta}_{\text{diag}}')_{ll} = (\boldsymbol{P}_{\text{left}})_{ll}(\boldsymbol{\theta}_{\text{diag}})_{ll}(\boldsymbol{P}_{\text{right}})_{ll}$. This results in $\boldsymbol{W}_1' = \boldsymbol{P}_1 \boldsymbol{W}_1$, $\boldsymbol{W}_l' = \boldsymbol{P}_l \boldsymbol{W}_l \boldsymbol{P}_{l-1}^T$ for $l \in \{2, \ldots, L-1\}$, and $\boldsymbol{W}_L' = \boldsymbol{I}_L \boldsymbol{W}_L \boldsymbol{P}_{L-1}^T = \boldsymbol{W}_L \boldsymbol{P}_{L-1}^T$.

It is important to note that the transformation matrices $\boldsymbol{S}_l$ are not fundamentally restricted to permutations. Many neural network components possess inherent symmetries richer than permutation symmetry, and modern architectures, such as Transformers, are often designed to effectively leverage such components. Recognizing and exploiting broader symmetry classes, when available, expands the set of valid reparameterization mappings. This, in turn, enhances alignment strategies by increasing the likelihood of discovering the low- or zero-barrier interpolation paths, thus allowing to achieve LMC.

For instance, Root Mean Square Layer Normalization (RMSNorm) component inherently possesses orthogonal symmetry (see Section 3.2). This property is particularly significant in architectures such as Transformers, where RMSNorm is commonly applied as a standalone operation—typically preceding major components like the attention or MLP blocks, without an immediately following element-wise activation. The absence of such a non-linearity preserves the richer orthogonal symmetry, which would otherwise be reduced to permutation symmetry. We can leverage such an architectural design to exploit the full orthogonal symmetry of RMSNorm for alignment purposes. We now examine the symmetry properties of RMSNorm in more detail.

Consider a network block defined as follows:

$$\boldsymbol{Y} = \boldsymbol{W}_1 \, \mathrm{RMSNorm}\big(\boldsymbol{W}_0 \, \boldsymbol{X} + \boldsymbol{b}; \, \beta, \gamma, \epsilon_N\big) = \boldsymbol{W}_1 \left( \gamma \, \frac{\boldsymbol{W}_0 \, \boldsymbol{X} + \boldsymbol{b}}{\sqrt{\frac{1}{N}\|\boldsymbol{W}_0 \, \boldsymbol{X} + \boldsymbol{b}\|_2^2 + \epsilon_N}} + \beta \right).$$

Introduce an orthogonal matrix $\boldsymbol{O} \in \mathbb{R}^{M \times N}$ ($M \geq N$, $\boldsymbol{O}^\top \boldsymbol{O} = \boldsymbol{I}$). Since $\|\boldsymbol{O}z\|_2 = \|z\|_2$, inserting $\boldsymbol{O}^\top \boldsymbol{O} = \boldsymbol{I}$ gives

$$\boldsymbol{Y} = \boldsymbol{W}_1 \, \boldsymbol{O}^\top \left( \gamma \, \frac{\boldsymbol{O}(\boldsymbol{W}_0 \, \boldsymbol{X} + \boldsymbol{b})}{\sqrt{\frac{1}{N}\|\boldsymbol{O}(\boldsymbol{W}_0 \, \boldsymbol{X} + \boldsymbol{b})\|_2^2 + \epsilon_N}} + \boldsymbol{O}\beta \right).$$

To express the normalization over $M$ elements, define $c = \sqrt{\frac{N}{M}}$, $\epsilon_M = \frac{N}{M}\epsilon_N$. Then

$$\frac{1}{\sqrt{\frac{1}{N}\|z\|^2 + \epsilon_N}} = c \, \frac{1}{\sqrt{\frac{1}{M}\|z\|^2 + \epsilon_M}},$$

so the block can be rewritten as

$$\boldsymbol{Y} = \boldsymbol{W}_1 \, \gamma \, \boldsymbol{O}^\top c \left( \frac{\boldsymbol{O}(\boldsymbol{W}_0 \, \boldsymbol{X} + \boldsymbol{b})}{\sqrt{\frac{1}{M}\|\boldsymbol{O}(\boldsymbol{W}_0 \, \boldsymbol{X} + \boldsymbol{b})\|_2^2 + \epsilon_M}} + \frac{1}{c}\frac{\boldsymbol{O}\beta}{\gamma} \right).$$

Hence the network remains functionally equivalent when written as

$$\boldsymbol{Y} = \boldsymbol{W}_1' \, \mathrm{RMSNorm}\big(\boldsymbol{W}_0' \, \boldsymbol{X} + \boldsymbol{b}'; \, \beta', \gamma', \epsilon_M\big), \tag{6}$$

with transformed parameters

$$\boldsymbol{W}_1' = \boldsymbol{W}_1 \, \gamma \, \boldsymbol{O}^\top c, \qquad \boldsymbol{W}_0' = \boldsymbol{O} \, \boldsymbol{W}_0, \qquad \boldsymbol{b}' = \boldsymbol{O} \, \boldsymbol{b}, \tag{7}$$

and normalization constants

$$\gamma' = \mathbf{1}_M, \qquad \beta' = \frac{1}{c}\frac{\boldsymbol{O}\,\beta}{\gamma} = \sqrt{\frac{M}{N}}\,\frac{\boldsymbol{O}\,\beta}{\gamma}. \tag{8}$$

This derivation for RMSNorm underscores a critical point: the nature of a component's inherent symmetries dictates the set of valid transformation matrices $\boldsymbol{S}_l$ that can be used for reparameterization while preserving its functionality. Let $\mathcal{S}_l$ denote the set of allowed transformation matrices for a given layer $l$ of dimension $d_l$. If a layer exclusively admits permutation symmetry, then $\mathcal{S}_l = \mathcal{P}_{d_l}$, the finite set of $d_l \times d_l$ permutation matrices. However, if a component, such as RMSNorm blocks in Transformers, exhibit orthogonal symmetry, the set of permissible transformations expands to $\mathcal{S}_l = \mathcal{O}(d_l)$, the orthogonal group. For components allowing even more general transformations, this could be $\mathcal{S}_l = \mathcal{GL}(d_l, \mathbb{R})$, the general linear group of invertible matrices. These sets of transformations are nested, with $\mathcal{P}_{d_l} \subset \mathcal{O}(d_l) \subset \mathcal{GL}(d_l, \mathbb{R})$, where $\mathcal{P}_{d_l}$ is a finite group, while $\mathcal{O}(d_l)$

and $\mathcal{GL}(d_l, \mathbb{R})$ are continuous, significantly larger Lie groups. The overall alignment transformation $\pi$ for the entire network is constructed by selecting an appropriate $\boldsymbol{S}_l \in \mathcal{S}_l$ for each layer or component. Consequently, by identifying and utilizing the richest symmetry class available for each network component—for example, employing transformations from $\mathcal{O}(d_l)$ for an RMSNorm layer rather than being restricted to the smaller set $\mathcal{P}_{d_l}$ (which might be the case if its orthogonal symmetry were immediately broken by a subsequent element-wise activation function)—we drastically expand the search space of valid, function-preserving reparameterizations $\pi(\boldsymbol{\theta})$. This significantly larger search space offers more degrees of freedom in the alignment process, thereby substantially increasing the potential to discover configurations $\boldsymbol{\theta}'$ (aligned parameter vector) that lie on low-loss linear paths and thus achieve Linear Mode Connectivity between independently trained models.

An additional important consequence of this symmetry is that, since the matrix $\boldsymbol{O}$ can be rectangular with $M \geq N$, the transformation can expand the dimensionality of the layer while preserving exact functional equivalence. In other words, *it is possible to increase the width of certain components* (such as RMSNorm layers and their connected blocks) *while maintaining their outputs unchanged*, thanks to the orthogonal symmetry. This enables reparameterizations that not only align models of identical architecture but also align models of differing widths—expanding the expressive power of alignment strategies. More generally, by exploiting symmetries such as $\mathcal{O}(d_l)$, we can define valid mappings $\pi(\boldsymbol{\theta})$ that traverse across architectures within a family of functionally equivalent models. This further enlarges the search space for alignment and opens the door to architecture-aware alignment methods and interpolation across model scales.

## F Discussion

**Geometry of Transformer minima.** Establishing zero-barrier LMC in Transformers reveals a surprisingly connected geometry in the loss landscape, where symmetries resolve apparent barriers and enable smooth interpolation between independently trained models. This extends prior observations for simpler architectures (e.g., MLPs and CNNs Draxler et al. [2018], Garipov et al. [2018]) while opening several avenues for future research. Below, we outline implications of particular importance.

**Loss landscape insights.** Our results support the view that Transformer solutions reside in broad, connected basins once functional symmetries (e.g., permutations and orthogonals) are accounted for, challenging the notion of isolated minima in high-dimensional parameter spaces. Prior work conjectured such connectivity primarily for SGD-trained MLPs and CNNs Entezari et al. [2021]; we empirically extend this to Transformers. Moreover, these findings complement the Lottery Ticket Hypothesis Frankle et al. [2020], which posits that dense networks contain sparse, trainable subnetworks ("winning tickets"). Our results suggest that such subnetworks may reside within the same connected regions, yielding more navigable landscapes for pruning, distillation, and efficient training from scratch.

**Federated and continual learning.** In federated learning, heterogeneous client models often hinder aggregation (e.g., under FedAvg). Symmetry-based alignment can pre-process weights to resolve these discrepancies, reducing variance and improving convergence under non-IID data. In continual learning, merging task-specific updates with previous model states preserves connectivity across sequential minima—barriers drop to zero post-alignment—offering a parameter-efficient alternative to replay buffers or regularization-based methods.

**Adversarial robustness directions.** Adversarially robust models are typically associated with flatter minima, which enhance generalization by enlarging basins of attraction Stutz et al. [2021]. However, adversarial training often incurs a trade-off between clean and robust accuracy. A promising extension of our framework could involve transporting a high-accuracy (non-robust) model into the basin of a robust counterpart via symmetry-aligned interpolation, potentially inheriting beneficial properties from both. While robustness is not evaluated here, the existence of zero-barrier paths provides a principled mechanism for such *basin hopping*, motivating future exploration.

**Role of positional encodings.** Positional encodings further modulate the symmetries admissible in language models. **Absolute Positional Encodings (APEs)**, as employed in our GPT-2 setup,

are additive at the embedding layer and do not alter internal block invariances: permutation matrices $P \in \mathcal{S}_1$ in feed-forward networks (FFNs), semi-permutation matrices $\tilde{P} \in \mathcal{S}_2$ in multi-head attention (MHA), and orthogonal matrices $O \in \mathcal{S}_3$ in residual connections remain valid symmetry elements. By contrast, **Rotary Positional Encodings (RoPE)** embed rotations directly within query–key projections, enforcing block-diagonal constraints on the residual orthogonals $O \in \mathcal{S}_3$. This decomposition breaks the full orthogonal group, necessitating specialized alignment procedures that respect these subgroup structures. Developing such RoPE-aware symmetry mappings is a promising direction for extending our approach to modern large-scale language models.

