# OpenReview forum: "Generalized Linear Mode Connectivity for Transformers"
_NeurIPS.cc/2025/Conference — NeurIPS 2025 oral_

### Official Review · Reviewer_My19 · 2025-07-01

**Clarity:** 3
**Significance:** 3
**Originality:** 3
**Rating:** 4
**Confidence:** 5

**Summary:**

This paper extends permutation symmetries in neural networks, as mostly discussed in the literature, to other 3 types of symmetries. Building on this symmetry framework they describe methods for aligning two independently trained Transformer models. These mathod range from weight matching, to activation matching and learned matching (with gradient).

**Questions:**

1) The proposed Learned matching is a reminiscnet of "Learning Neural Network Subspaces" by Wortsman et al. What are the diifferences, highlighting novelty?
2) One important motivations to remove barriers is to have them averaged to improved accuracy. Did you try to make ensembles given the zero barrier in CIDARs?
3) This is a great start, but only limited to ViTs to CIFARs for zero barriers. How about more complex tasks, as stated in literature, ImageNet?
4) experimental design: Which ViTs did you use (B/L, etc) and how well the models are optimized for CIFAR tasks? what is the top1 accuracy? and how important it is to have a overfit model on the downstream task?
5) Given the promising results of learned matching, did you try to remove the barrier for two non-ViT based models on ImageNet? We need some pointers to compare the effectiveness of the best method here this with some baselines.
6) What do you think of applying learned matching to DiTs?

**Ethical Concerns:**

["NO or VERY MINOR ethics concerns only"]

**Final Justification:**

After reading the discussions and responses to my questions/concerns I increase my score to 4. I hope that this line of work motivate future works on applications on generative DiTs too.

**Limitations:**

- some of the claims are not clear to me: Is this work the first to look into matching of ViTs or the first to make the barrier zero in CIFARs?
- experiments are limited to proof of concepts like ViT on CIFARs and one GPT
- some other limitations include (that are also correctly reflected in Appendix): For example the paper does not provide results for more than two models, and I wonder why.
- remaining are my notes in the questions

**Paper Formatting Concerns:**

no concern

**Quality:**

3

**Strengths And Weaknesses:**

+ a fundamental problem yet important direction to improve current SOTA models
+ well written, structured and formulated, easy to understand and follow

- limited experiments and results (more in questions)

---

> ### Author Rebuttal · Authors · 2025-07-30
>
> We thank the reviewer for their thoughtful and constructive feedback. We appreciate the recognition of our work's foundational importance and clear writing. Below, we address the reviewer's questions and limitations point by point, incorporating new experimental evidence where relevant to strengthen our responses.
>
> We agree with the reviewer that the paper can only benefit from more extensive experiments. For this reason, we have added experiments on *Tiny-ImageNet*, and for merging *more than two models* with each other, achieving *zero-loss barriers* in both settings (see response to *Q3*). We commit to include even larger-scale results (e.g., on ImageNet-1k) in the camera-ready version.
>
> We hope these additions clarify and strengthen the paper. If they address the reviewer's concerns, we would appreciate the reviewer's consideration in updating their score.
>
> Before delving into the reviewer's valuable questions and remarks, we would like to shortly add some context to our obtained results.
>
> ## Context on results
> We emphasize that our core contribution is foundational: prior to our work, **Linear Mode Connectivity (LMC) had not been demonstrated for Transformer architectures — not even on simple benchmarks like MNIST**. Even for MLPs and CNNs, which have been the primary architectures for studying LMC, achieving zero-loss paths often required substantial architectural modifications. For example, Git Re-Basin [1] demonstrates LMC for ResNet-20 on CIFAR-10 only after widening the model by a factor of 20.
>
> In contrast, our work is the first to demonstrate *zero-loss* linear interpolation between independently trained Vision Transformers and GPT-2 models — without any width expansion or architectural changes. **This is a striking and somewhat surprising result**: despite their greater depth and complexity, standard Transformers exhibit LMC even more readily than CNNs. This provides the first direct evidence that LMC extends to modern Transformer-based architectures.
>
> We agree that evaluating on larger-scale models and datasets like CLIP or ImageNet would strengthen our claims. At the same time, we believe that starting with benchmarks like CIFAR and Tiny ImageNet is essential: prior to our work, LMC had not been observed in Transformers at *any* scale. These controlled settings allow us to rigorously establish this phenomenon and provide a solid foundation for scaling to more complex models.
>
> ## Questions
> **Q1: Difference between "Learning Neural Network Subspaces" by Wortsman et al. and our method.**
>
> We understand how these studies might appear similar at first glance, as both involve low-loss subspace. To clarify, the key distinctions are in their goals and technical approaches:
>
> * **Our paper**: Our learned matching is a *post-hoc alignment technique* for merging independently trained models. Our objective is to learn *function-preserving transformations to connect existing models with low-loss linear paths*, keeping the original weights frozen. This enables the analysis of the geometry of loss landscapes and the connectedness of Transformers.
> * **Wortsman et al. [2]**: Their method is a *training procedure* that optimizes an entire subspace (e.g., a line of models) from scratch during a single training run. It updates the boundary model weights by backpropagating losses from sampled subspace points, aiming to produce robust models or cheap ensembles—*not to align pre-trained models*.
>
> **Q2:** We thank the reviewer for the insightful comment. While our primary focus is on LMC in Transformers, we agree that the behavior of interpolated models is of interest. We typically observe reduced test loss at the interpolation midpoint, but no improvement in test accuracy. For instance, on CIFAR-10, ViTs achieve ~85% accuracy and ~0.55 loss, whereas the midpoint interpolated model has ~84% accuracy and ~0.49 loss. This suggests improved calibration, which is supported by notably higher Expected Calibration Error (ECE) scores in interpolated models.
>
> **Q3: Extension to more complex tasks.**
>
> We agree that extending our experiments to more challenging datasets and merging settings enhances the paper's contributions. In response, we include the following new results:
>
> * **Tiny-ImageNet**: We train ViTs on Tiny-ImageNet from scratch, using a cosine learning rate schedule with linear warmup, achieving test accuracy of ~44.6%. The merged model achieves a **zero-loss barrier**, and a test accuracy of 44.57%, demonstrating that independently trained ViTs on Tiny-ImageNet converge to the same loss basin. Loss landscape plots and interpolation figures will be added to the camera-ready version.
>
> * **Multi-model merging**: We extend our experiments to merging **three models simultaneously**. Rather than sampling a single interpolation coefficient at random, we now sample two coefficients from a Dirichlet distribution. We evaluate this setting for ViTs on CIFAR-10, CIFAR-100, and Tiny-ImageNet, and discover zero-loss barrier at the midpoint $\tfrac{1}{3} \Theta_A + \tfrac{1}{3} \Theta_B + \tfrac{1}{3} \Theta_C$.
>
> **Q4: Experimental setup of ViTs.**
> We did not use standard ViT-B/L variants, as our study requires training from scratch with multiple random initializations to probe loss landscape connectivity. Instead, we used custom ViTs tailored for CIFAR to allow controlled, repeatable experiments.
>
> * **Architecture details**: The full architectural configuration is included in Appendix A.1 of the paper. Our ViTs use: patch size of 4, embedding dimension of 256, MLP dimension of 512, 6 layers, and 8 attention heads.
>
> * **Optimization and Performance**: The models are well-optimized, achieving ~85% top-1 accuracy on CIFAR-10. This indicates effective learning without severe under- or overfitting—no significant train-test gap was observed. These results align with prior work on similar custom ViTs.
>
> **Q5: Removing barriers for non-ViT based models.**
> We appreciate the interest in extending our method beyond Transformers. Our core claim is that Transformers exhibit richer symmetries than earlier architectures—extending beyond simple permutations. While permutation-based methods (e.g., Git Rebasin) suffice for CNNs like ResNets, they fail in Transformers, where normalization layers and attention mechanisms introduce orthogonal and invertible symmetries. These are absent in CNNs, where nonlinearities like ReLUs break such structure. Consequently, applying our framework to CNNs reduces to permutation-only alignment—already well-explored in prior work we build upon.
>
> Nevertheless, these richer symmetries are not entirely unique to Transformers. For example, **MAMBA** exhibits orthogonal invariance along its residual path, highlighting an exciting direction for future work on linear mode connectivity in MAMBA.
>
> **Q6: Learned matching for DiTs.** We thank the reviewer for this forward-looking question. Diffusion Transformers (DiTs) share core components with the ViTs and GPT-2 models we studied, including Transformer blocks with multi-head attention, residual connections, and normalization layers. Our identified symmetries and approach to align them should directly translate to DiTs.
>
> It would be highly interesting to see results for interpolated DiTs trained on different datasets (e.g., landscapes vs. portraits). This could lead to **hybrid generations** — mixing styles or domains in novel ways.
>
> ## Limitations
> **L1: Unclear claims.** We are the **first method to achieve zero- and low-loss barriers between independently trained Transformers**. We are not the first paper to look into matching Transformers. For example, Verma et al. [3] match Transformers by permuting neurons according to their activations, referred to as "activation matching" in our Methods and Results chapter.
>
> ## References
> [1] Ainsworth et al. (2023), *Git Re-Basin: Merging Models modulo Permutation Symmetries*.
>
> [2] Wortsman et al. (2021), *Learning Neural Network Subspaces*.
>
> [3] Verma & Elbayad (2024), *Merging Text Transformer Models from Different Initializations*.

---

> > ### Comment · Reviewer_My19 · 2025-08-05
> > **after rebuttal**
> >
> > I appreciate the efforts made by authors to address my questions and concerns, that also overlap with other reviewers. I am still interested to see the extension of these work to DiTs and their effectiveness in model merging.
> >  Given the reponses and also positive feedback from other reviewers I would like to increase my score to 4.

---

> > > ### Author Response · Authors · 2025-08-08
> > > **Thank you!**
> > >
> > > We sincerely thank the reviewer for their thoughtful and constructive feedback, and for your updated score. We are glad to hear that our responses have addressed your questions and concerns, and we greatly appreciate your positive assessment of our work.
> > >
> > > We are especially excited about the reviewer’s interest in extending our approach to *Diffusion Transformers (DiTs)*. As mentioned, we believe this is a promising direction, and we look forward to exploring this in future work.
> > >
> > > *Thank you for your time, careful review, and valuable input*. We look forward to presenting the final version of the paper, with all the improvements based on your feedback.

---

### Official Review · Reviewer_HV2f · 2025-07-01

**Clarity:** 4
**Significance:** 4
**Originality:** 3
**Rating:** 5
**Confidence:** 5

**Summary:**

The paper presents a thorough accounting of the (invariant) symmetries present in modern transformer architectures. These go beyond the permutations symmetries found in convolutions networks and include richer mathematical structures, including orthogonal and invertible transforms. Using a carefully constructed optimizer, the authors are able to apply symmetries to bring two models into the same “loss basin,” exhibiting zero loss barrier between the models for the first time.

**Questions:**

1. For language models, how do common forms of positional encoding affect the symmetries?
2. This is subjective, but I find the title to be a bit misleading. It is not the connectivity that is being generalized but rather the symmetries which enable that connectivity. This is an important distinction and I think the current title undersells the papers results.

**Ethical Concerns:**

["NO or VERY MINOR ethics concerns only"]

**Final Justification:**

This work is the first to demonstrate zero barrier for independently trained transformers, which is a substantial achievement. Additionally the presentation of the symmetries in the paper is excellent and I believe it will be of great resource to those interested in transformers.

**Limitations:**

The main limitation (along with weakness 1) is that the loss barrier persists in the GPT-2 results. This may suggest that either the enumeration of symmetries is incomplete for those models, that symmetries are insufficient to give linear mode connectivity, or that the optimization procedure is not finding a good solution. Some discussion here would be very useful to indicate directions for future improvement.

**Quality:**

3

**Strengths And Weaknesses:**

## Strengths

1. The presentation of the transformer symmetries is really excellent.
2. The achievement of zero loss barrier for transformers is significant. This is a natural next step following the near zero barrier results in Git Re-Basin [1]. However it has been several years and this paper is the first to actually achieve the result!

[1] Ainsworth, Samuel, Jonathan Hayase, and Siddhartha Srinivasa. "Git Re-Basin: Merging Models modulo Permutation Symmetries." The Eleventh International Conference on Learning Representations.

## Weaknesses

1. The best results are obtained by directly optimizing over the various symmetries to minimize the barrier. This is (potentially) computationally expensive and it is not clear to me when this expense would be justified.
2. Limited evaluation. The main ViT results study one architecture and two datasets (CIFAR 10 and 100). It would be interesting to see merges for more difficult datasets such as ImageNet or CLIP models, where ViTs are commonly used.

---

> ### Author Rebuttal · Authors · 2025-07-30
>
> We thank the reviewer for their thoughtful review and positive assessment of our work's presentation and significance. We are pleased that the reviewer recognizes the value in our demonstration of zero-loss barriers for Transformers. Below, we address the points raised, incorporating clarifications and new experimental evidence, where relevant.
>
> ## Weaknesses
>
> **W1: Computational costs.**
> We understand the reviewer's concern about the cost incurred by our Learned Matching (LM) method, which optimizes over symmetries to minimize barriers. This approach does involve optimization over symmetries to minimize the barrier, which is more computationally intensive than data-free methods like Weight Matching (WM).
>
> * **The relationship between model heterogeneity and alignment difficulty.** Our primary aim is to demonstrate linear mode connectivity in Transformers for the first time. We acknowledge the reviewer’s concern about practical applicability. There is indeed a tradeoff: **the more divergent the models, the harder they are to align and merge**. In our case, the models were trained independently from different random initializations and seeds, making alignment nontrivial without learning it. However, such divergence is less pronounced in practical settings. For instance, in **federated learning**, models are typically trained from the same initialization and aggregated periodically. Weight matching could mitigate divergence in this context, enabling less frequent aggregation and thus faster training.
>
> * **Understanding the gap between weight- and learned matching.** A first step towards bridging the gap between weight- and learned matching is understanding how their alignment solutions differ. Interestingly, we observe that the only difference between the solutions of weight- and learned matching is the orthogonal matrix that aligns the residual path. Thus, the key to unveiling zero-loss paths without requiring any data or finetuning is better alignment of the residual path. We will add this ablation study to the camera-ready version.
>
> **W2: Limited evaluation:** Our work provides the **first demonstration of Linear Mode Connectivity (LMC) in standard Transformer architectures** - a foundational result that we believe constitutes a meaningful contribution in its own right. Notably, prior work has struggled to achieve zero-barrier interpolation even in simpler settings (e.g., Git Rebasin only achieves zero barrier by radically increasing layer widths on ResNets trained on CIFAR). That said, we agree that additional empirical validation would only further strengthen our claims.
>
> * **Tiny-ImageNet**: We train Vision Transformers (ViTs) from scratch on Tiny-ImageNet, employing a cosine learning rate decay and linear warmup. Our models attain a test accuracy of approximately 44.6%. When we merge two independently trained models, we observe a **barrier-free interpolation path** in the loss landscape, with the merged model achieving 44.57% test accuracy—essentially matching the original models. This suggests that the independently trained ViTs converge into the same optimization basin. We plan to include visualizations (loss surface and interpolation curves) in the camera-ready version.
>
> * **Merging more than two models**: We further explore merging **three models at once** by sampling two interpolation weights from a Dirichlet distribution. This extends our study to a broader merging space. We apply this approach to ViTs trained on CIFAR-10, CIFAR-100, and Tiny-ImageNet, and observe that even the centroid point $\tfrac{1}{3} \Theta_A + \tfrac{1}{3} \Theta_B + \tfrac{1}{3} \Theta_C$ remains on a **zero-loss path**, reinforcing our finding that ViT training trajectories consistently arrive in a shared solution basin.
>
> We commit to including larger-scale ImageNet-1k experiments and a broader range of architectures in the camera-ready version, to further confirm our framework's applicability.
>
> ## Questions
>
> **Q1: Positional encodings and symmetries:**  We thank the reviewer for this question, which raises an interesting point worthy of analysis. We will include additional pointers in the paper, as these may help inform future research.
>
> - **Absolute Positional Encodings (APEs):** As used in our GPT-2 setup, these are added at the input embedding layer and do not impact the internal symmetries of Transformer blocks (e.g., permutations in FFNs, semi-permutations in attention, or orthogonals in residuals).
> - **Rotary Positional Encodings (RoPE):** These embed rotations directly into attention, likely imposing new constraints like block-diagonal structures on orthogonal symmetries.
>
> We are excited to add a section on the symmetry of different positional embeddings in the camera-ready version.
>
> **Q2: Title Feedback:**  We value the reviewer's perspective that the title could better spotlight the role of richer symmetries in enabling LMC between Transformers - this was indeed our goal but our title choice may not have been entirely on point. To address this, we are considering revisions such as:
> - *Generalized Symmetries Enable Linear Mode Connectivity in Transformers*
> - *Unlocking Linear Mode Connectivity in Transformers with Richer Symmetries*
> - *Beyond Permutations: Achieving Linear Mode Connectivity in Transformers with Richer Symmetries*
> These aim to more precisely convey our contributions while maintaining focus on linear mode connectivity.
>
> ## Limitations
> We thank the reviewer for noting the contrast between ViTs and GPT-2 regarding Linear Mode Connectivity. This difference aligns with findings from Juneja et al. [1], who show that NLP models often converge to functionally distinct basins, even with the same initialization and training data. These basins correspond to different generalization strategies, and interpolating between them leads to loss barriers, breaking LMC. In contrast, vision models such as ViTs tend to generalize in more homogeneous ways, which facilitates the emergence of LMC. Our observations support this interpretation. We will add a Discussion section to the camera-ready version addressing this.
>
> ## References
> [1] Juneja et al. (2023), *Linear Connectivity Reveals Generalization Strategies*.

---

> ### Comment · Reviewer_HV2f · 2025-08-03
>
> I would like to thank the authors for their thorough response. The additional results sound very promising and the proposed additions sound like they would further improve the robustness of the results and provide more useful information for other researchers.
>
> Regarding the title, I would rather not make a subjective judgement regarding which is best, but I do think all three address my concern with the original title, so the authors clearly understood my original hesitation.
>
> In light of this (assuming the promised changes are made), I am increasing my score to 5.

---

> > ### Author Response · Authors · 2025-08-08
> > **Thank you!**
> >
> > We sincerely thank the reviewer for their thoughtful and constructive feedback, and for recognizing the significance of this work. As the reviewer points out, it has been many years since Git Re-Basin [1] achieved near-zero barriers for MLPs and CNN-based architectures, and approaches adapting this methodology for Transformers have failed to achieve similar results [2]. We believe our work marks a crucial step forward in the LMC community by demonstrating that exploiting symmetries beyond permutations enables the discovery of low-loss paths in modern architectures such as Transformers.
> >
> > The reviewer’s insightful comments have been invaluable in helping us improve the manuscript. We are particularly excited to include a chapter on the symmetry of different positional encodings and to explore the discrepancy between vision and text-based models in the context of LMC, which we believe will further enhance the impact of our findings.
> >
> > Once again, *thank you for your time, careful review, and valuable input*. We look forward to presenting the final version of the paper, incorporating your suggestions.
> >
> > ## References
> > [1] Ainsworth et al. (2023), *Git Re-Basin: Merging Models modulo Permutation Symmetries*.
> >
> > [2] Verma and Elbayad (2024), *Merging Text Transformer Models from Different Initializations*.

---

### Official Review · Reviewer_DXBv · 2025-07-02

**Clarity:** 3
**Significance:** 4
**Originality:** 3
**Rating:** 5
**Confidence:** 3

**Summary:**

The paper discusses linear mode connectivity and introduces a unified framework capturing richer symmetries than in the present literature, including permutations, semi-permutations, orthogonal transformations, and invertible maps that explain these connections and enable discovery of smooth linear paths between ViTs and GPT-2 models. These findings highlight the importance of symmetry-aware analysis for understanding neural network model space geometry.

**Questions:**

Please see weaknesses. I'm ready to revise my score once these are carefully addressed.

**Ethical Concerns:**

["NO or VERY MINOR ethics concerns only"]

**Final Justification:**

The authors have addressed my concerns in their rebuttal. I believe the paper provides a solid contribution to the literature and have revised my score accordingly.

**Limitations:**

yes

**Quality:**

3

**Strengths And Weaknesses:**

Strengths:
- The authors do a great job carefully analyzing different layers and activation functions. They show low / zero-loss linear interpolation paths between independently trained ViTs and GPT-2 models, establishing deeper connectivity in Transformer loss landscapes than what has been known so far.
- The proposed unified treatment of permutations, semi-permutations, orthogonal, and general invertible maps goes beyond prior permutation-only approaches, capturing richer symmetries in modern architectures. The paper explores multiple alignment strategies (activation, weight, and learned matching), with learned matching demonstrating strong empirical performance by fully eliminating interpolation barriers in experiments.
- Results on CIFAR-10, CIFAR-100, and BookCorpus empirically support the main claims.

Weaknesses:
- The paper reports the barrier in terms of the test loss. It would also be interesting to see the corresponding accuracy / perplexity (in particular, how much is 0.41 loss barrier between two GPT-2 networks?)
- Add missing citations, e.g., the definition of LMC, the methods such as activation matching and weight matching, etc. Please be generous as to where specific concepts and methods were introduced, as you build on those in your work.
- Algorithm 1 implements a straight-through estimator. Please cite the literature following a similar approach. How does your approach compare to existing STE-based methods? What new insights or technical contributions does your broader symmetry framework introduce beyond extending to more transformation classes? How does the proposed method compare to OT-fusion?
- The reviewer is missing a well thought discussion about the implication of LMC between transformer networks. What do the proposed (very impressive) findings enable?

---

> ### Author Rebuttal · Authors · 2025-07-31
>
> We thank the reviewer for their thorough and constructive feedback. We appreciate the recognition of our contributions, including the unified symmetry framework and the empirical demonstration of zero-loss paths in Transformers. We agree that contextualizing results with performance metrics, adding citations, clarifying comparisons, and discussing implications will strengthen the paper. We will revise accordingly, incorporating these changes into the camera-ready version.
>
> Before delving into the reviewer's insightful questions, we would first like to highlight some additional results:
>
> ## Additional Results
> * **Tiny-ImageNet**: We trained Vision Transformers (ViTs) from scratch on Tiny-ImageNet, utilizing a cosine learning rate schedule with a linear warmup phase. The resulting models reach a test accuracy close to 44.6%. When interpolating between two such models, the combined model achieves 44.57% test accuracy and shows a **smooth, barrier-free loss trajectory**. This supports the idea that separately trained ViTs on Tiny-ImageNet converge to a shared region in parameter space. We will include loss landscape and interpolation visualizations in the final version.
>
> * **Merging three models**: We expand our experiments to include **simultaneous merging of three models**. To do this, we sample two mixing coefficients from a Dirichlet distribution, allowing us to explore convex combinations beyond pairwise interpolation. This method is evaluated on ViTs trained on CIFAR-10, CIFAR-100, and Tiny-ImageNet. Notably, the interpolated model at the equal-weight point $\tfrac{1}{3} \Theta_A + \tfrac{1}{3} \Theta_B + \tfrac{1}{3} \Theta_C$ also lies on a **zero-barrier path**, indicating that all three models share a common low-loss region in the optimization landscape.
>
> ## Weaknesses
>
> **W1: Contextualizing loss barriers with accuracy and perplexity.** We appreciate the reviewer’s feedback and agree that the paper would benefit from clearer presentation of model performance. On CIFAR-10, the individual trained models achieve a test accuracy of approximately 85%, with a corresponding test loss around 0.55. The interpolated model at the midpoint exhibits a slightly lower test accuracy of ~84%, but a reduced test loss of ~0.49. This decrease in test loss, despite similar accuracy, suggests **improved calibration** — a finding supported by the interpolated model’s higher Expected Calibration Error (ECE) score.
>
> For GPT-2, the baseline model achieves a test loss of approximately 3.55, corresponding to a perplexity of 34.8. In comparison, the merged model shows a slightly higher test loss of ~3.91, resulting in a perplexity of 51.9.
>
> We will include detailed performance metrics for both individual and merged models in the camera-ready version, along with calibration plots that highlight the improved calibration observed in interpolated models.
>
> **W2: Add citations.** We thank the reviewer and we agree on the importance of crediting prior work and will extend our citations accordingly. Currently, we cite Frankle et al. [1] for LMC; Ainsworth et al. [2] for Git Re-Basin; Tatro et al. [3] for activation matching; Sing & Jaggi [4] for weight-matching/OT-Fusion; Verma et al. [5] for the first work on exploring LMC for Transformers; Ashkboos et al. [6] for uncovering orthogonal symmetries in Transformers.
>
> **W3: Relation to STE-based Methods and OT-Fusion.**  We agree that it is important to provide a clear distinction from STE-based methods, including foundational STE work.
>
> * **STE Comparison**: Our Algorithm 1 uses STE for backpropagation through soft latent matrices, similar to Git Re-Basin. However, our core contribution is not in how permutation matrices are learned, but in *extending beyond permutations* to identify and leverage broader *transformer symmetries* for *function-preserving alignment*. While Git Re-Basin [2] and Verma et al. [3] align residual weights via permutations (using OT or STE), we go further by aligning with *orthogonal transformations* derived via SVD. This richer symmetry perspective reveals *zero-loss barriers* between independently trained Transformers—unreachable with permutations alone. We empirically validate this: restricting to permutation learning via STE (as in Git Re-Basin’s “Learned matching (permutations)”) consistently leads to high loss barriers (see Tables 2 and 3), underscoring the need for richer symmetry exploitation.
>
> * **OT-Fusion Comparison**: OT-Fusion [4] introduced activation- and weight-matching to align MLPs and CNNs, but extending these techniques to Transformers is non-trivial. Verma et al. [3] adapt OT-Fusion’s *activation matching* to Transformers, a method we include in our results as “activation matching,” which our method consistently outperforms (Tables 2 and 3). OT-Fusion’s weight matching resembles Git Re-Basin’s first iteration. In Appendix B.2, we evaluate Git Re-Basin’s multi-iteration version (“permutation matching” in Figure 5) and find it performs significantly worse than our method, failing to reduce loss barriers after the first iteration.
>
> **W4: Implications of LMC in Transformers.** We thank the reviewer for highlighting the need to discuss implications more thoroughly — our results indeed uncover profound insights into Transformer landscapes. We will add a discussion section, incorporating these points for greater clarity and impact.
>
> Establishing zero-barrier LMC in Transformers reveals a surprisingly connected geometry in the loss landscape, where symmetries resolve apparent barriers, allowing smooth interpolation between independently trained models. This extends prior observations from simpler architectures (e.g., MLPs and CNNs, as in Draxler et al. [7] and Garipov et al. [8]) while opening exciting avenues for future research. Below, we outline the key implications with enhanced technical detail:
>
> **Loss Landscape Insights:** These results support the view that Transformer solutions reside in broad, connected basins once functional symmetries (e.g., permutations and orthogonals) are accounted for, challenging the notion of isolated minima in high-dimensional spaces. Prior work conjectured such connectivity primarily for SGD-trained solutions in fully connected or convolutional networks (e.g., Entezari et al. [9]); we empirically extend this to Transformers. Furthermore, this complements the Lottery Ticket Hypothesis [1], which posits that dense networks contain sparse, trainable subnetworks ("winning tickets"); our LMC findings suggest these tickets may lie within the same connected regions, enabling more navigable landscapes for pruning, distillation, and efficient training from scratch.
>
> **Federated and Continual Learning:** In federated learning, heterogeneous client models often hinder aggregation (e.g., via FedAvg); alignment algorithms could preprocess weights to resolve symmetries, reducing variance and improving convergence on non-IID data. For continual learning, merging task-specific updates with prior states mitigates catastrophic forgetting by preserving connectivity across sequential minima, as barriers drop to zero post-alignment—offering a parameter-efficient alternative to replay buffers or regularization methods.
>
> **Adversarial Robustness Directions:** Adversarially robust models are associated with flatter minima, which enhance generalization by enlarging basins of attraction (Stutz et al. [10]). However, adversarial training often trades off standard accuracy for robustness. A compelling extension of our work could involve transporting a high-accuracy (non-robust) model into the basin of a robust counterpart via symmetry-aligned interpolation, potentially inheriting benefits from both (e.g., improved clean and adversarial accuracy). While we do not evaluate robustness empirically here, the zero-barrier paths provide a foundational mechanism for such "basin hopping," inspiring future investigations.
>
> ## References
>
> [1] Frankle et al. (2020), *Linear Mode Connectivity and the Lottery Ticket Hypothesis*.
>
> [2] Ainsworth et al. (2023), *Git Re-Basin: Merging Models modulo Permutation Symmetries*.
>
> [3] Tatro et al. (2020), *Optimizing Mode Connectivity via Neuron Alignment*.
>
> [4] Singh and Jaggi (2023), *Model Fusion via Optimal Transport*.
>
> [5] Verma and Elbayad (2024), *Merging Text Transformer Models from Different Initializations*.
>
> [6] Ashkboos et al. (2024), *SliceGPT: Compress Large Language Models by Deleting Rows and Columns*.
>
> [7] Draxler et al. (2018), *Essentially No Barriers in Neural Network Energy Landscape*.
>
> [8] Garipov et al. (2018), *Loss Surfaces, Mode Connectivity, and Fast Ensembling of DNNs*.
>
> [9] Entezari et al. (2021), *The Role of Permutation Invariance in Linear Mode Connectivity of Neural Networks*.
>
> [10] Stutz et al. (2021), *Relating Adversarially Robust Generalization to Flat Minima*.

---

> > ### Comment · Reviewer_DXBv · 2025-08-03
> >
> > The authors have addressed my concerns in their rebuttal. I believe the paper provides a solid contribution to the literature and have revised my score accordingly.

---

> ### Author Response · Authors · 2025-08-07
> **Thank you!**
>
> We sincerely thank the reviewer for their thoughtful and constructive feedback, and we are grateful for the updated score, particularly for **raising the significance rating to 4**. We are thrilled that the reviewer recognizes the high significance of this paper, especially for the LMC and model merging community. We are excited to further clarify the relationship between our method and prior approaches, including OT-Fusion and STE-based methods, in the final manuscript, as we have done in this rebuttal.
>
> Once again, we thank the reviewer for their time, and valuable suggestions, all of which have greatly contributed to improving the quality of this work. We are actively available should any further questions or clarifications arise.

---

### Official Review · Reviewer_nSkR · 2025-07-06

**Clarity:** 3
**Significance:** 3
**Originality:** 3
**Rating:** 4
**Confidence:** 2

**Summary:**

Previous studies on linear mode connectivity (LMC) in deep neural networks have primarily focused on symmetries arising from neuron permutations. This limits the applicability of these studies to modern architectures, such as Transformers. To address this, the authors propose a unified framework covering four classes of symmetries: permutations, semi-permutations, orthogonal transformations, and general invertible maps. This framework generalizes existing approaches. Using this framework, the authors demonstrate the existence of low- or zero-barrier linear interpolation paths between Vision Transformer and GPT-2 models that were trained independently.

**Questions:**

- Is there a specific reason for selecting the interpolation coefficient $\lambda$ from the interval $U(0.4, 0.6)$? For instance, would a wider range such as $U(0.3, 0.7)$ also be feasible?

**Ethical Concerns:**

["NO or VERY MINOR ethics concerns only"]

**Final Justification:**

I find the results showing that LMC exists in Transformer architectures interesting. I appreciate that the authors addressed my question adequately. I would like to see more diverse experiments included in the camera-ready version. I will maintain my original score.

**Limitations:**

Yes

**Quality:**

3

**Strengths And Weaknesses:**

**Strengths**

- The overall writing is clear and easy to follow.
- The diagrams in the paper are clearly presented and easy to understand.
- The newly proposed symmetry classes are novel and are presented with clear mathematical formulations.
- The authors empirically demonstrate the existence of linear mode connectivity with sufficiently low loss barriers in Transformer architectures.

**Weaknesses**

The main limitation of the paper is that the empirical results are not sufficiently extensive.

- The experiments on Vision Transformers are primarily conducted on relatively simple datasets such as CIFAR-10/100, and the GPT-2 experiments are limited to the comparatively small BookCorpus dataset. Additional evaluation on more complex datasets, where Transformers are more commonly applied, would help validate the generality of the findings.
- The experiments are restricted to ViT and GPT-2 models. Including results on a broader range of Transformer architectures would further support the practicality and general applicability of the proposed method.

---

> ### Author Rebuttal · Authors · 2025-07-31
>
> We thank the reviewer for their positive and constructive feedback. We appreciate the recognition of our framework's novelty, the clarity of the presentation, and the significance of our findings.
> Our work provides the **first demonstration of Linear Mode Connectivity (LMC) in standard Transformer architectures** - a foundational result that we believe constitutes a meaningful contribution in its own right. Notably, prior work has struggled to achieve zero-barrier interpolation even in simpler settings (e.g., Git Rebasin only achieves zero barrier by increasing layer widths on convolutional networks trained on CIFAR). That said, we fully agree that additional empirical validation would further strengthen our claims. We have already conducted new experiments and will include more extensive large-scale results in the camera-ready version.
> ### **Additional Results**
> * **Tiny-ImageNet**: We train Vision Transformers (ViTs) from the ground up on Tiny-ImageNet using a cosine decay schedule with linear warmup. The resulting models achieve around 44.6% test accuracy. Upon merging two separately trained models, we observe no increase in test loss—achieving 44.57% accuracy at the interpolation midpoint. This **absence of a loss barrier** indicates that the independently trained ViTs converge into the same region of the loss landscape. Plots illustrating the interpolation behavior and loss surfaces will be provided in the camera-ready version.
>
> * **Three-way model merging**: We also generalize our analysis to merging **three models simultaneously**. In this setting, two interpolation coefficients are drawn from a Dirichlet distribution to explore the interior of the simplex. We evaluate this approach on ViTs trained on CIFAR-10, CIFAR-100, and Tiny-ImageNet, and find that the merged model at the uniform average point $\tfrac{1}{3} \Theta_A + \tfrac{1}{3} \Theta_B + \tfrac{1}{3} \Theta_C$ continues to lie on a **barrier-free path**, further confirming that these models reach a common solution basin during training.
>
> We will incorporate experiments on the larger-scale ImageNet-1k dataset, along with a more diverse set of architectures, in the camera-ready version to further validate the generality of our framework.
>
> ## Questions
> **Q1: Is there a specific reason for selecting the interpolation coefficient \$\lambda\$ from the interval $\[0.4, 0.6]\$? For instance, would a wider range such as $\[0.1, 0.9]\$ also be feasible?**
> This touches on an important aspect of our optimization strategy.
> Our choice to sample \$\lambda\$ from the narrower $\[0.4, 0.6]\$ interval is deliberate. Empirically, the loss barrier tends to peak around the midpoint of the interpolation path (i.e., \$\lambda \approx 0.5\$). By focusing our optimization on this most challenging region, we are able to more effectively reduce the **maximum barrier height**, which is the critical quantity in establishing LMC.
>
> While a broader range such as $\[0.1, 0.9]\$ is technically feasible, it would divert optimization efforts away from the region that most directly contributes to barrier elimination. Indeed, in Figure 5 of the Appendix we investigate the performance of different sampling strategies, including a broader range of $\[0.0, 1.0]\$, which we empirically find to deteriorate performance around the midpoint. Thus, our narrower focus provides a better efficiency–effectiveness trade-off.

---

> > ### Comment · Reviewer_nSkR · 2025-08-03
> >
> > Thank you to the authors for taking the time to write the rebuttal.
> > I find the results showing that LMC exists in Transformer architectures interesting.
> > I appreciate that the authors addressed my question adequately.
> > I would like to see more diverse experiments included in the camera-ready version.
> > I will maintain my original score.

---

> > > ### Author Response · Authors · 2025-08-07
> > > **Thank you!**
> > >
> > > We sincerely thank the reviewer for the insightful comments and for engaging with our rebuttal. We are very pleased to hear that the reviewer acknowledges the significance of our findings and that we were able to adequately address your review.
> > >
> > > Regarding the question of our coefficient sampling strategy, we would like to further clarify that in Figure 5 of the appendix, we have experimented with other sampling strategies, such as sampling from a Gaussian-like distribution centered at the midpoint, and fixing $\lambda$ at 0.5. For the newly added experiment demonstrating linear mode connectivity between *three independently trained Transformers*, we also conducted ablation studies with respect to the sampling strategy, including sampling from Dirichlet distributions like $(10,10,10)$, $(5,5,5)$, and others. Our findings indicate that distributions closer to the midpoint, such as $(10,10,10)$, perform best.
> > >
> > > Once again, we thank the reviewer for strengthening our paper with their valuable insights. We remain actively available to address any further questions or clarifications.

---

### Comment · Area_Chair_4AMm · 2025-08-06
**[General Reminder for Authors and Reviewers] Author-Reviewer Discussion Phase Ending Soon**

Dear Authors and Reviewers,

As you know, the deadline for author-reviewer discussions has been extended to August 8. If you haven’t done so already, please ensure there are sufficient discussions for both the submission and the rebuttal.

Reviewers, please make sure you complete the mandatory acknowledgment **AND** respond to the authors’ rebuttal, as requested in the email from the program chairs.

Authors, if you feel that any results need to be discussed and clarified, please notify the reviewer. Be concise about the issue you want to discuss.

Your AC

---

### Note · Authors · 2025-08-16

We want to sincerely thank the Area Chairs and reviewers for their thoughtful engagement and valuable feedback, which have further strengthened this work. We are very pleased that *all reviewers recommend acceptance and acknowledge the significance of our contributions*.

This paper establishes, for the first time, **zero-barrier linear mode connectivity in Transformer architectures**, a milestone that has remained elusive despite years of effort since Git Re-Basin demonstrated low-barrier paths only for CNNs/MLPs under *widened* architectures. Our unified symmetry framework (permutations, semi-permutations, orthogonals, and general invertibles) explains why prior permutation-only approaches fail in Transformers and provides the tools to resolve these barriers.

Reviewers raised requests for more experiments, which we addressed with new results on Tiny-ImageNet, and multi-model alignment (three-way model interpolation), *both exhibiting barrier-free paths*. We will include ImageNet-1k results and additional architectures in the camera-ready version.

We believe these findings extend the LMC line of research to modern architectures such as Transformers, enabling principled weight-space alignment for applications such as model merging, federated, and continual learning. We look forward to sharing the final version with the NeurIPS community.

---

### Decision · Program_Chairs · 2025-09-17

**Decision:**

Accept (oral)

**Comment:**

The recommendation is based on the reviewers' comments, the area chair's evaluation, and the author-reviewer discussion.

This paper studies linear mode connectivity in transformer models, including vision transformers and GPT-2 models, by considering a broader class of weight symmetry. All reviewers find the studied setting novel and the results provide new insights. The authors’ rebuttal has successfully addressed the major concerns of reviewers.

In the post-rebuttal phase, all reviewers were satisfied with the authors’ responses and agreed on the decision of acceptance. Overall, I recommend acceptance of this submission. I also expect the authors to include the new results and suggested changes during the rebuttal phase in the final version.

Given that this work provides a solid demonstration of the generalized linear model connectivity in transformer models, I believe the insights to understanding the geometry of the loss landscape of transformer models and the associated applications are significant, and I recommed for an oral presentation.